# Loss of H3K9 trimethylation alters chromosome compaction and transcription factor retention during mitosis

Dounia Djeghloul [1] ✉, Andrew Dimond [1], Sherry Cheriyamkunnel[1], Holger Kramer [2], Bhavik Patel[3], Karen Brown[1], Alex Montoya [2], Chad Whilding[4], Yi-Fang Wang[5], Matthias E. Futschik[5], Nicolas Veland [1], Thomas Montavon[6], Thomas Jenuwein [6], Matthias Merkenschlager[7] & Amanda G. Fisher [1] ✉

Recent studies have shown that repressive chromatin machinery, including DNA methyltransferases and polycomb repressor complexes, binds to chromosomes throughout mitosis and their depletion results in increased chromosome size. In the present study, we show that enzymes that catalyze H3K9 methylation, such as Suv39h1, Suv39h2, G9a and Glp, are also retained on mitotic chromosomes. Surprisingly, however, mutants lacking histone 3 lysine 9 trimethylation (H3K9me3) have unusually small and compact mitotic chromosomes associated with increased histone H3 phospho Ser10 (H3S10ph) and H3K27me3 levels. Chromosome size and centromere compaction in these mutants were rescued by providing exogenous first protein lysine methyltransferase Suv39h1 or inhibiting Ezh2 activity. Quantitative proteomic comparisons of native mitotic chromosomes isolated from wild-type versus Suv39h1/Suv39h2 double-null mouse embryonic stem cells revealed that H3K9me3 was essential for the efficient retention of bookmarking factors such as Esrrb. These results highlight an unexpected role for repressive heterochromatin domains in preserving transcription factor binding through mitosis and underscore the importance of H3K9me3 for sustaining chromosome architecture and epigenetic memory during cell division.

Heterochromatin and euchromatin are defined cytologically as condensed and decondensed regions of the genome, respectively[1,2]. Constitutive heterochromatin, although gene poor, contains noncoding DNA repeat elements that are often abundantly transcribed[3,4]. During interphase, heterochromatin-containing domains within different chromosomes cluster together forming structures termed chromocenters[5–8]. In mouse cells these dynamic structures are characteristically marked by a high density of histone 3 lysine 9 trimethylation

[1]Epigenetic Memory Group, MRC London Institute of Medical Sciences, Imperial College London, London, UK. [2]Biological Mass Spectrometry and Proteomics Facility, MRC London Institute of Medical Sciences, Imperial College London, London, UK. [3]Flow Cytometry Facility, MRC London Institute of Medical Sciences, Imperial College London, London, UK. [4]Microscopy Facility, MRC London Institute of Medical Sciences, Imperial College London, London, UK. [5]Bioinformatics, MRC London Institute of Medical Sciences, Imperial College London, London, UK. [6]Max-Planck Institute of Immunobiology and Epigenetics, Freiburg, Germany. [7]Lymphocyte Development Group, MRC London Institute of Medical Sciences, Imperial College London, London, UK. ✉e-mail: d.djeghloul@lms.mrc.ac.uk; amanda.fisher@lms.mrc.ac.uk

(H3K9me3) and histone histone 4 lysine 20 trimethylation, as well as the H3K9-associated heterochromatin protein 1 (HP1α or Cbx5)[1,7]. As chromosomes condense and enter mitosis, primary constrictions first become evident within the heterochromatin domains that correspond to centromeric microsatellite arrays, where the mitotic spindles will eventually bind. These centromeric regions are flanked by much larger domains of pericentric noncoding major satellite repeats which, in the mouse, account for approximately 3.6% of the genome[9–11].

The Suv39h1 and Suv39h2 histone lysine methyltransferases are hallmark enzymes of mammalian heterochromatin that catalyze the trimethylation of histone H3 at lysine 9 (H3K9)[12–14]. They form part of a larger group of enzymes capable of modifying H3K9 that includes G9a (Ehmt2), Glp (Ehmt1)[15,16], Setdb1 and Setdb2 (refs. [17,18]), which primarily mediate H3K9 mono- and dimethylation, as well as members of the Kdm1, Kdm3 and Kdm4 families of histone demethylases[19–23]. Although heterochromatin regulation is recognized as being essential in preserving nuclear architecture, genome stability and DNA repair, and for silencing transposon expression in early mouse development[24–27], the underlying repetitiveness of satellite DNA means that factors binding to heterochromatin are often excluded from conventional analyses.

In the present study we set out to examine the impact of repressive H3K9me3 on mitotic chromosome architecture and on the factors that bind chromosomes through mitosis. Several DNA-binding transcription factors (TFs) have been shown to remain bound to chromosomes during mitosis, including Foxa1, Esrrb, Sox2 and Gata1 (refs. [28–33]), where they occupy a subset of the genomic sites that are present during the interphase. Such demonstrations have raised the interesting possibility that 'mitotic bookmarking' by factors such as these could help to convey cellular identity to newly divided daughter cells. Recent studies have also shown that many components of repressive chromatin machinery, including those that characterize constitutive heterochromatin, are also retained at mitotic chromosomes during cell division[34–36]. Although this has prompted speculation of an interplay between mitotic booking factors and repressive chromatin states[37], there is currently a paucity of direct evidence to support this. In the present study, we confirm that factors mediating H3K9 trimethylation are indeed retained on native mitotic chromosomes isolated from different cell types. H3K9me3 depletion results in altered chromosome architecture, compensatory changes in the level and distribution of repressive H3K27me3 and discrete and reversible changes in the retention of specific mitotic bookmarking factors.

## Results

### Suv39h dn mitotic chromosomes are more compact

Previous studies had suggested that Suv39h1, Suv39h2 and HP1 proteins Cbx5 (HP1α), Cbx1 (HP1β) and Cbx3 (HP1γ), as well as other repressive chromatin machinery, remain bound to chromosomes during mitosis[34–36,38–40]. Suv39h1, Suv39h2 and HP1 enrichment at mitotic chromosomes was independently confirmed herein using established proteomic approaches[36] in mouse embryonic stem cells (ESCs) and mouse embryonic fibroblasts (MEFs) (outlined in Supplementary

Fig. 1a,b and depicted in Supplementary Fig. 1c). To investigate the impact of H3K9me3 on mitotic chromosome architecture, native mitotic chromosomes were isolated directly from wild-type (WT) ESCs (Fig. 1a–c) or MEFs (Fig. 1d–f) and compared with mitotic chromosomes isolated from cells lacking both Suv39h1 and Suv39h2 (Suv39h double null or *Suv39h dn*)[13,26]. To enable this, dividing cell cultures of ESCs were arrested in metaphase using demecolcine, incubated with polyamine buffer[36], and the resulting chromosomes stained with Hoechst 33258 and chromomycin A3. Individual chromosomes were isolated by flow cytometry as described previously[36] (Supplementary Fig. 1a provides a schematic of the approach and Supplementary Fig. 1b illustrates the gating strategy). Using purified preparations of chromosomes 19 and X as exemplars, mitotic chromosomes isolated from *Suv39h dn* ESCs[26] were shown to be smaller than their WT counterparts, and a marked increase in compaction at centromeric domains was noted (Fig. 1b,c).

To determine whether reduced size and enhanced condensation of mitotic chromosomes deficient in H3K9me3 were seen in other cell types, including differentiated cells, mitotic chromosomes from MEFs lacking Suv39h1 and Suv39h2 (ref. [13]) were also examined (Fig. 1d). Consistent with results obtained in ESCs, mitotic chromosomes isolated from these fibroblasts were smaller than WT equivalents and showed increased compaction at centromeres (Fig. 1e,f). To exclude these changes being an artifact of the isolation procedure, we also examined conventional metaphase spreads where chromosome-specific probes and DNA–fluorescence in situ hybridization (FISH) was used to label chromosomes 19 and X. These analyses confirmed that mitotic chromosomes from *Suv39h dn* ESCs were reproducibly smaller than their WT counterparts (Fig. 1g,h). To examine whether deficits in H3K9 mono- and dimethylation affect chromosome compaction, we isolated mitotic chromosomes from a panel of MEFs that lacked Suv39h1 and Suv39h2 (clustered regularly interspaced short palindromic repeats (CRISPR) clone B1) or other H3K9 methyltransferases, specifically deletions of both Setdb1 and Setdb2 (CRISPR clone A4 + 4-hydroxytamoxifen (4-OHT)), or of both G9a and Glp (CRISPR clone H7)[41] (Supplementary Fig. 1d). Analysis of isolated native mitotic chromosomes from clones A4 + 4-OHT, H7 and B1 fibroblasts versus WT controls showed that reduced chromosome size was a unique feature of cells lacking H3K9me3 (ref. [41]) (Supplementary Fig. 1e).

The compaction of mitotic chromosomes lacking H3K9me3 was unexpected because previous studies had shown that an absence of other repressive chromatin modifiers, such as DNA methylation or polycomb repressive complex 2 (PRC2) activity, produced chromosomes that were larger and more decondensed than equivalent WT mitotic controls[36]. To investigate the possibility that additional heterochromatin modifications, such as increased H3K27me3, might compensate for deficits in H3K9me3, we examined the distribution of modified histones in *Suv39h dn* and WT ESCs. As anticipated, H3K9me3 decorated centromeric domains of normal mitotic chromosomes (Fig. 2a and Supplementary Fig. 2a, green, WT top panel), but was absent from equivalent *Suv39h dn* chromosomes (Fig. 2a and Supplementary Fig. 2a, lower panel, quantified in the graph, left). Instead, we detected

**Fig. 1 | Suv39h1/Suv39h2 deficiency generates small and compact mitotic chromosomes. a–f**, Flow sorting and size measurements of chromosomes (Chr) 19 and X from WT or *Suv39h dn* mouse ESCs and MEFs. Flow karyotype of mitotic chromosomes is isolated from WT or *Suv39h dn* ESCs (**a**) and MEFs (**d**); gates used to isolate chromosomes 19 and X are indicated. Representative images (right) of mitotic chromosomes 19 (**b** and **e**) and X (**c** and **f**) from WT or *Suv39h dn* ESCs (**b** and **c**) and MEFs (**e** and **f**) are shown, where DAPI stain (gray) and Cenpa label (green) indicate the chromosome body and centromere, respectively. Scale bar, 5 μm. Boxplots (left of images) show area measurements of individual chromosomes and centromeres for WT and *Suv39h dn* cells. Minimum, lower quartile, median, upper quartile and maximum values are indicated (*n* = minimum 100 chromosomes analyzed for each cell line over 3 independent experiments). *P* values of statistically significant changes, measured by unpaired, two-tailed Student's *t*-tests, are indicated. **g,h**, Representative images of WT or *Suv39h dn* ESC metaphase spreads stained with either chromosome 19 painting probe (green) (**g**) or chromosome X painting probe (green) (**h**), in addition to γSat probe (red) and DAPI (blue). Scale bars, 4 μm and 1 μm for the metaphase spread and zoom-in images, respectively. Chromosome and centromere sizes of chromosomes 19 and X were calculated on metaphase spreads of WT and *Suv39h dn* ESCs. Boxplots show chromosome and centromere area measurements for WT and *Suv39h dn* spreads. Minimum, lower quartile, median, upper quartile and maximum values are indicated (*n* = minimum 25 chromosomes analyzed on metaphase spreads for each line over 3 independent experiments). **b,c,e–h**, *P* values of statistically significant changes, measured by unpaired, two-tailed Student's *t*-tests, are indicated. Source data, including the precise numbers of chromosomes analyzed, are provided.

increased levels of H3K27me3 across centromeric domains of *Suv39h dn* chromosomes (Fig. 2b and Supplementary Fig. 2b, pink). Ingress of H3K27me3 at centromeres was consistent with previous studies showing increased H3K27me3 at chromocenters in interphase in the absence of H3K9me3 (refs.[42–44]). We also detected significant increases in histone H3 phospho Ser10 (H3S10ph) on *Suv39h dn* metaphase

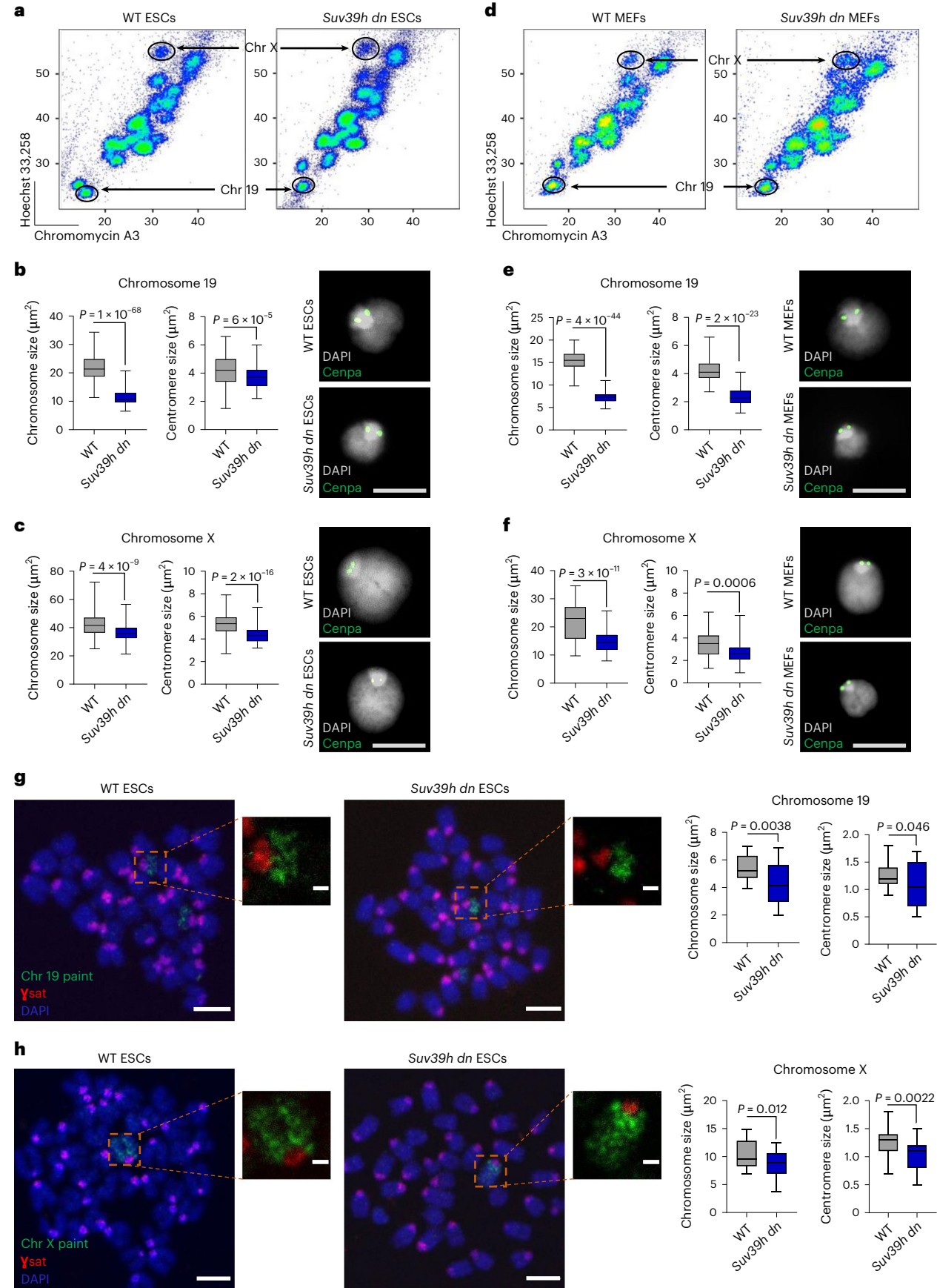

chromosomes (Fig. 2c, yellow) compared with WT equivalents. Increased H3K27me3 levels and ingress of this mark at centromeric domains were also seen in mitotic chromosomes isolated from *Suv39h dn* MEFs (Supplementary Fig. 2c), but not on equivalent chromosomes isolated from either *Setdb1/Setdb2* double null or *G9a/Glp* double null mutant cells (Supplementary Fig. 2c).

## PRC2 inhibition restores *Suv39h dn* mitotic chromosome size

To investigate whether enhanced compaction of *Suv39h dn* mitotic chromosomes was the result of compensatory increases in H3K27me3 rather than loss of H3K9me3 as such, *Suv39h dn* ESCs were treated for 48 hours (h) with drugs that either inhibit PRC2 activity (GSK343, an Ezh2 methyltransferase inhibitor) or block DNA methylation (5-azacytidine (5-Aza)) (Fig. 2d, schematic). Pretreatment with GSK343 reduced global H3K27me3 levels on mitotic chromosomes 19 (Fig. 2e, upper panel) and X (Fig. 2f, upper panel), and effectively restored chromosome size and centromere compaction to that of WT equivalents (Fig. 2e,f, lower panels). Pretreatment of *Suv39h dn* ESCs with 5-Aza resulted in a smaller increase in mitotic chromosome size and centromere decompaction. In WT ESCs, treatment with either drug resulted in modest increases in mitotic chromosome size (Supplementary Fig. 2d). Collectively these data show that the compact structure of *Suv39h dn* mitotic chromosomes can be alleviated by inhibiting the activity of PRC2. To exclude chromatin accessibility being grossly altered in *Suv39h dn* samples, assay for transposase-accessible chromatin using sequencing (ATAC-seq) was performed on WT and *Suv39h dn* asynchronous ESCs, mitotic-arrested ESCs, as well as sorted ESC mitotic chromosomes. Global ATAC-seq profiles of *Suv39h dn* and WT chromosomes indicated that they were broadly comparable, as illustrated for chromosome 19 (Fig. 2g) and as shown by differential accessibility analysis (Supplementary Fig. 2e). This suggests that changes in chromosome-scale compaction observed in *Suv39h dn* ESCs are independent of local changes in chromatin accessibility. We also examined mitotic progression and cell cycle in ESCs lacking *Suv39h1/h2* activity using live-cell imaging[45] and propidium iodide staining. As shown in Supplementary Fig. 2f,g, dividing cultures of *Suv39h dn* ESCs produced a similar proportion of cells in G1, S and G2/M phases of the cell cycle because their WT counterparts and mitotic duration were also comparable (Supplementary Fig. 2g, boxplot, right panel). These data exclude any delay in mitotic progression as the cause of increased mitotic chromosome compaction in *Suv39h dn* cells.

## Proteomic analysis of *Suv39h dn* mitotic chromosomes

To analyze the factors binding to unfixed (native) metaphase chromosomes in *Suv39h dn* and WT cells (MEFs and ESCs), we performed proteomic analyses using liquid chromatography–tandem mass spectrometry (LC-MS/MS) in which the data were analyzed using the label-free quantification (LFQ) algorithm in the MaxQuant platform, as detailed previously[36]. For these experiments, an equivalent number of metaphase chromosomes ($10^7$) in samples pre- and post-chromosome sorting were compared (schematically shown in Supplementary Fig. 1a), in at least three biological replicates loaded in technical duplicates on the LC-MS/MS (Supplementary Fig. 3a–d, Supplementary Data 1 and 2). After filtering raw MaxQuant data in Perseus as described in Methods, a comparable number of protein hits were identified in WT and *Suv39h dn* MEF samples (Supplementary Data 1). Among these hits, 4,468 were detected in both mitotic lysate and sorted chromosomes for WT MEFs and 4519 were detected in both mitotic lysate and sorted chromosomes for *Suv39h dn* MEFs (Supplementary Data 1). To identify proteins that were enriched on mitotic chromosomes (red), depleted (blue) or unaltered in abundance between mitotic lysate pellets and purified chromosomes (gray), data were subjected to multiple Student's *t*-tests with permutation-based false discovery rate (FDR) (detailed in Methods) and displayed as volcano plots (Fig. 3a). Both WT and *Suv39h dn* MEFs showed similar proportions of proteins enriched on sorted mitotic chromosomes (11.9% and 16.3%, respectively). Comparing these enriched hits between WT and *Suv39h dn* MEFs (based on the corresponding protein IDs) revealed that many of these chromosome-bound proteins were common to both genotypes (464 of 848), although a subset of candidates (287 of 848) was not enriched in *Suv39h dn* MEF samples compared with WT (Fig. 3b, listed in Supplementary Data 1). These proteins clustered in function, being associated with processes such as transcription, chromosome organization, cell cycle, development and nucleosome organization (Fig. 3c), in addition to H3K9 trimethylation. No obvious functional group was seen among the small group of factors (97 of 848) that were preferentially detected on *Suv39h dn* MEF mitotic chromosomes, compared with WT.

We performed an analogous comparison in ESCs (Fig. 3d–f). After filtering, 4,990 protein hits were detected in both mitotic lysate and sorted chromosome samples for WT ESCs and 4,521 were detected in both mitotic lysate and sorted chromosomes for *Suv39h dn* ESCs (Supplementary Data 2). Of these, 15.6 and 13.5% were identified as being significantly enriched on mitotic chromosomes for WT and *Suv39h dn* ESCs, respectively (Fig. 3d). Although a majority (56%) of chromosome-bound candidates were common to both genotypes, a subset (287 of 920) was not detected in the absence of both Suv39h enzymes (Fig. 3e). These candidates showed gene ontology (GO) term assignments that were remarkably similar to those identified previously in MEFs, encompassing chromosome organization, transcription, DNA repair, cellular responses to stress, chromatin silencing and H3K9 trimethylation (compare Fig. 3c,f, Supplementary Data 1 and 2). Taken together, these data highlight a commonality in the functional pathways affected by H3K9me3 removal in very divergent cell types: pluripotent stem cells and differentiated fibroblasts.

---

**Fig. 2 | *Suv39h dn* mitotic chromosomes show elevated levels of H3K27me3 and H3S10ph and can be resized by inhibiting PRC2 activity. a,b,** Representative images (right) of immunofluorescence labeling of histone H3K9me3 (**a**) (green) or histone H3K27me3 (**b**) (pink) on chromosome 19 isolated from WT or *Suv39h dn* ESCs, where DAPI counterstain is shown in light gray. Scale bar, 5 μm. Plots (left of the images) show H3K9me3 (**a**) or H3K27me3 (**b**) mean intensities, measured at centromeric regions. **c**, Representative images of immunofluorescence labeling of histone H3S10ph (yellow) on WT and *Suv39h dn* metaphase-arrested ESCs, where DAPI counterstain is shown in blue. Scale bar, 4 μm. H3S10ph mean intensities were measured on mitotic chromosomes for each condition (*n* = minimum 40 chromosomes analyzed for each cell line over 3 independent experiments (**a**–**c**). *P* values of statistically significant changes, measured by unpaired, two-tailed Student's *t*-tests, are indicated. **d**, Experimental strategy used to measure mitotic chromosome size of *Suv39h dn* ESCs after treatment with DNA methylation or PRC2 inhibitors (5-Aza or GSK343, respectively). **e,f**, Representative images of immunofluorescence labeling of histone H3K27me3 (red) on mouse chromosome 19 (**e**) and chromosome X (**f**) isolated from *Suv39h dn* ESCs pretreated with DMSO, 5-Aza or GSK343. DAPI counterstain is shown in light gray. Scale bar, 5 μm. H3K27me3 mean intensities were measured at centromeric regions and whole chromosomes; the mean ± s.d. is shown (*n* = minimum 50 chromosomes analyzed over 3 independent experiments). *P* values of statistically significant decreases compared with DMSO treatment, measured by unpaired, two-tailed Student's *t*-tests, are indicated. Boxplots show area measurements of individual chromosomes and centromeres for each condition. Minimum, lower quartile, median, upper quartile and maximum values are indicated (*n* = minimum 100 chromosomes analyzed for each condition over 3 independent experiments). *P* values of statistically significant increases compared with DMSO treatment, measured by unpaired, two-tailed Student's *t*-tests, are indicated. **g**, Chromatin accessibility profile across chromosome 19 for WT and *Suv39h dn* asynchronous (Asynch.) and mitotic ESCs and flow-sorted mitotic chromosomes, shown as $\log_2$(enrichment of ATAC-seq signal). **a**–**c**,**e**,**f**, Source data, including the precise numbers of chromosomes analyzed, are provided.

**Altered retention of TFs on *Suv39h dn* mitotic chromosomes**

To further investigate the factors that require H3K9me3 to remain efficiently bound to mitotic chromosomes in dividing ESCs, we looked in detail at the representation of pluripotency-associated proteins in sorted mitotic chromosomes from WT and *Suv39h dn* ESC samples.

Factors such as Sox2, Utf1, Zfp296, Sall4, Dppa2 and Dppa4, which we previously identified as being bound to unfixed mitotic ESC chromosomes[36], showed an equivalent representation in control and mutant samples (Fig. 4a). In the absence of Suv39h enzymes and H3K9me3, however, factors such as Esrrb (estrogen-related receptor-β, a

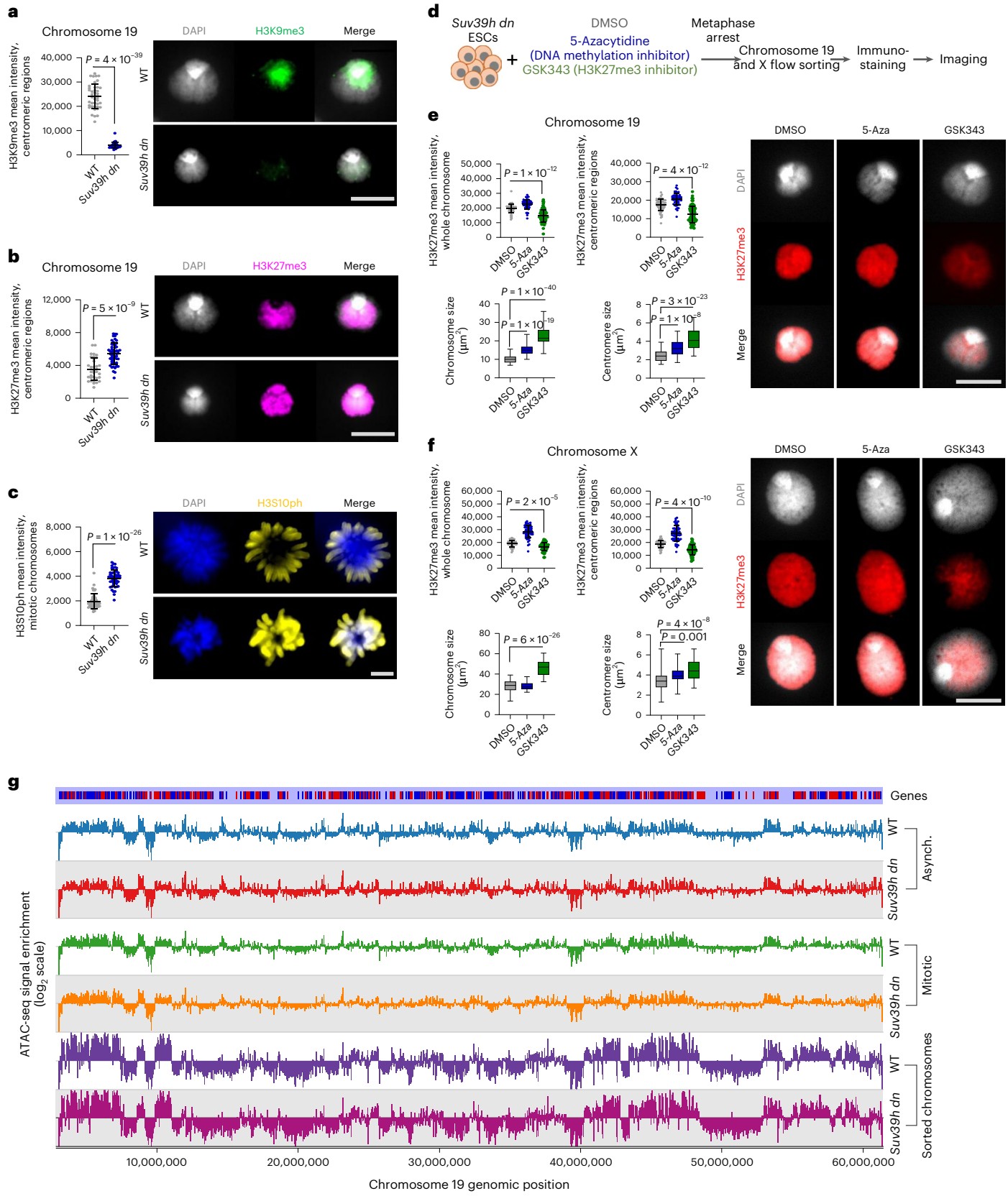

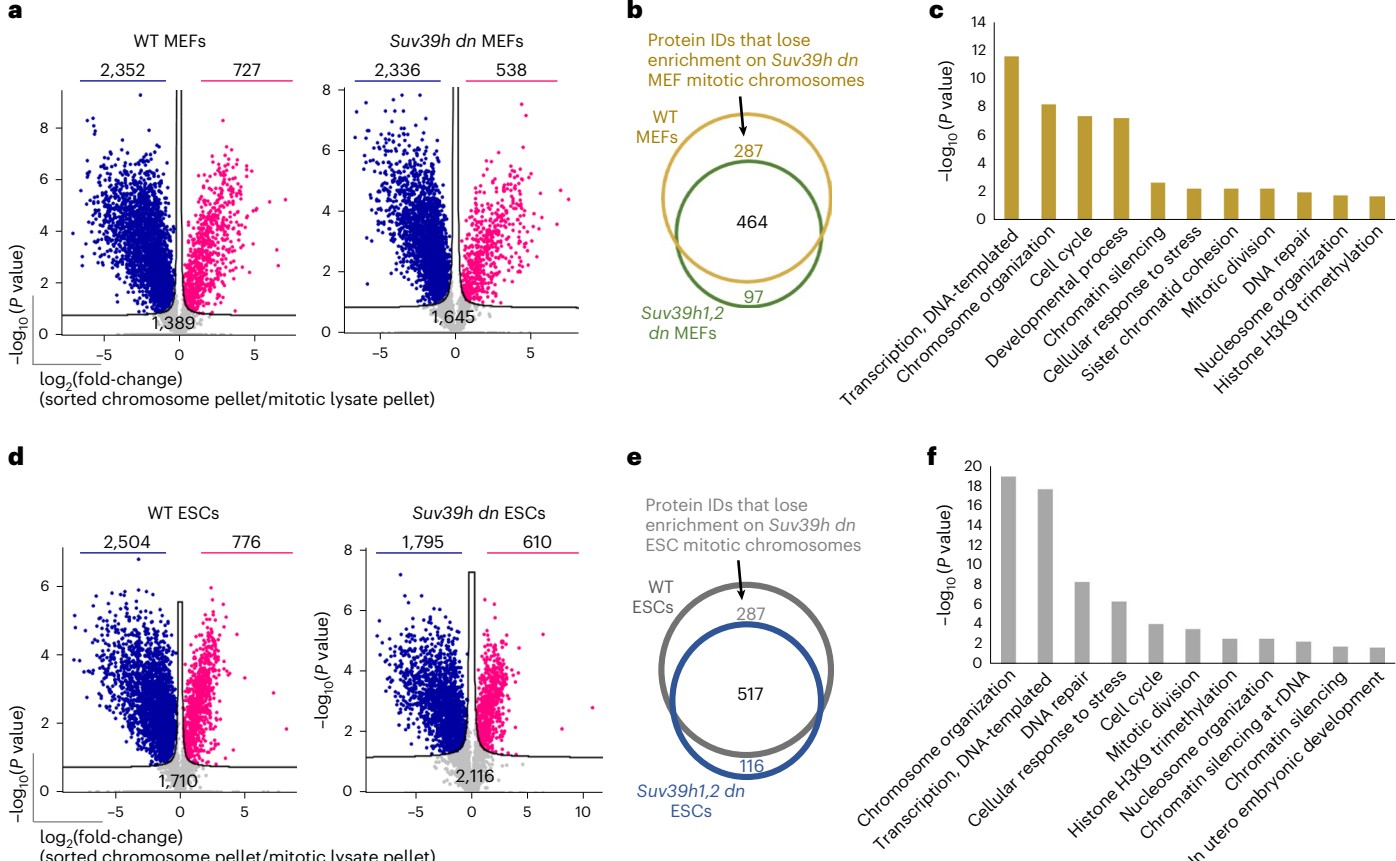

**Fig. 3 | Proteomic analysis reveals a cadre of chromosome-binding factors that require H3K9me3 to remain efficiently bound during mitosis. a**, Volcano plots of proteins significantly enriched (red), depleted (blue) or not significantly changed (gray) on sorted chromosomes relative to mitotic lysate pellets for WT (left) and *Suv39h dn* (right) MEFs (unpaired, two-tailed Student's *t*-test, permutation-based FDR < 0.05, s0 = 0.1 (*n* = 3 independent experiments each measured in technical duplicate; see Methods for details). Proteins were plotted as log₂(fold-change LFQ intensity of sorted chromosome pellets/LFQ intensity of mitotic lysate pellet versus significance) (−log₁₀(*P*)) using Perseus software). The number of proteins in each category is indicated on the volcano plot. **b**, Venn diagram showing the overlap of protein IDs enriched on mitotic chromosomes between WT and *Suv39h dn* MEF samples. **c**, GO term (biological process) analysis of proteins that lose enrichment on *Suv39h dn* mitotic MEF chromosomes compared with WT. Analysis was performed at http://geneontology.org using Fisher's exact test with FDR correction. **d**, Volcano plots (as in **a**) of proteins significantly enriched (red), depleted (blue) or not significantly changed (gray) on sorted chromosomes relative to mitotic lysate pellets for WT (left) and *Suv39h dn* (right) mouse ESCs. **e**, Venn diagram showing the overlap of protein IDs enriched on mitotic chromosomes between WT and *Suv39h dn* mouse ESC samples. **f**, GO term (biological process) analysis (as in **c**) of proteins that lose enrichment on *Suv39h dn* mitotic mouse ESC chromosomes compared with WT.

well-characterized mitotic bookmarking factor[31,32]), Tead4 (TEA domain transcription factor 4) and Tbx3 (T-box transcription factor) were no longer enriched on mitotic chromosomes (Fig. 4a). To exclude such deficits being simply the result of lower levels of expression by mutant cells, we compared our proteomic data with previously published transcriptomic data[27]. For both genotypes, there was a strong correlation between protein abundance in mitotic lysates and chromosome samples and between protein and transcript levels (Supplementary Fig. 4a). However, differences in the proteome of *Suv39h dn* mitotic chromosomes, relative to WT, appeared independent of differences in the mitotic lysate or underlying transcriptome (Supplementary Fig. 4b). This suggests that a loss of TF retention in *Suv39h dn* mitotic chromosomes is unlikely to be explained by changes in their abundance. To explore this in more detail, we examined the retention of a specific candidate, Esrrb. In WT and *Suv39h dn* ESCs, Esrrb levels were broadly comparable (Supplementary Fig. 4c) and proteomic data confirmed that Esrrb was equivalently represented in WT and *Suv39h dn* mitotic lysates (Supplementary Fig. 4d). However, sorted mitotic chromosomes from *Suv39h dn* ESCs showed an underrepresentation of Esrrb (Supplementary Fig. 4e) compared with equivalent WT controls.

To validate this result, we isolated native mitotic chromosomes 19 and X from *Suv39h dn* and WT ESCs and analyzed Esrrb levels by immunofluorescence. As shown in Fig. 4b, Esrrb (red) was significantly reduced in *Suv39h dn* samples compared with WT ones. This decreased retention of Esrrb was not accompanied by any pronounced loss of chromatin accessibility at Esrrb bookmarking sites[32], as judged by ATAC-seq analysis of asynchronous, mitotic and sorted chromosome preparations (Supplementary Fig. 4f). This is consistent with a global similarity in ATAC-seq data derived from WT and *Suv39h dn* samples (Fig. 2g and Supplementary Fig. 2e). However, direct analyses of Esrrb binding by chromatin immunoprecipitation (ChIP), using a double fixation protocol that preserves DNA–TF interactions in mitotic samples[31–33], revealed deficits in Esrrb bookmarking in *Suv39h dn* ESCs (Fig. 4c). We selected candidate sites from previous publications[31] as representing sites that were either bookmarked throughout mitosis by Esrrb (*Capn2*, *Esrrb*, *Jam2-s1*, *Jam2-s2*, *Tbx3* and *Tet2*), bound by Esrrb only in interphase (*Mgat3* and *Twistnb*) or negative control sites that do not bind Esrrb (*Esrrb-3′* and *Actb*)[31]. A statistically significant decrease in Esrrb binding was observed at five of seven bookmarked sites analyzed in *Suv39h dn* compared with WT mitotic chromosome samples (upper panel).

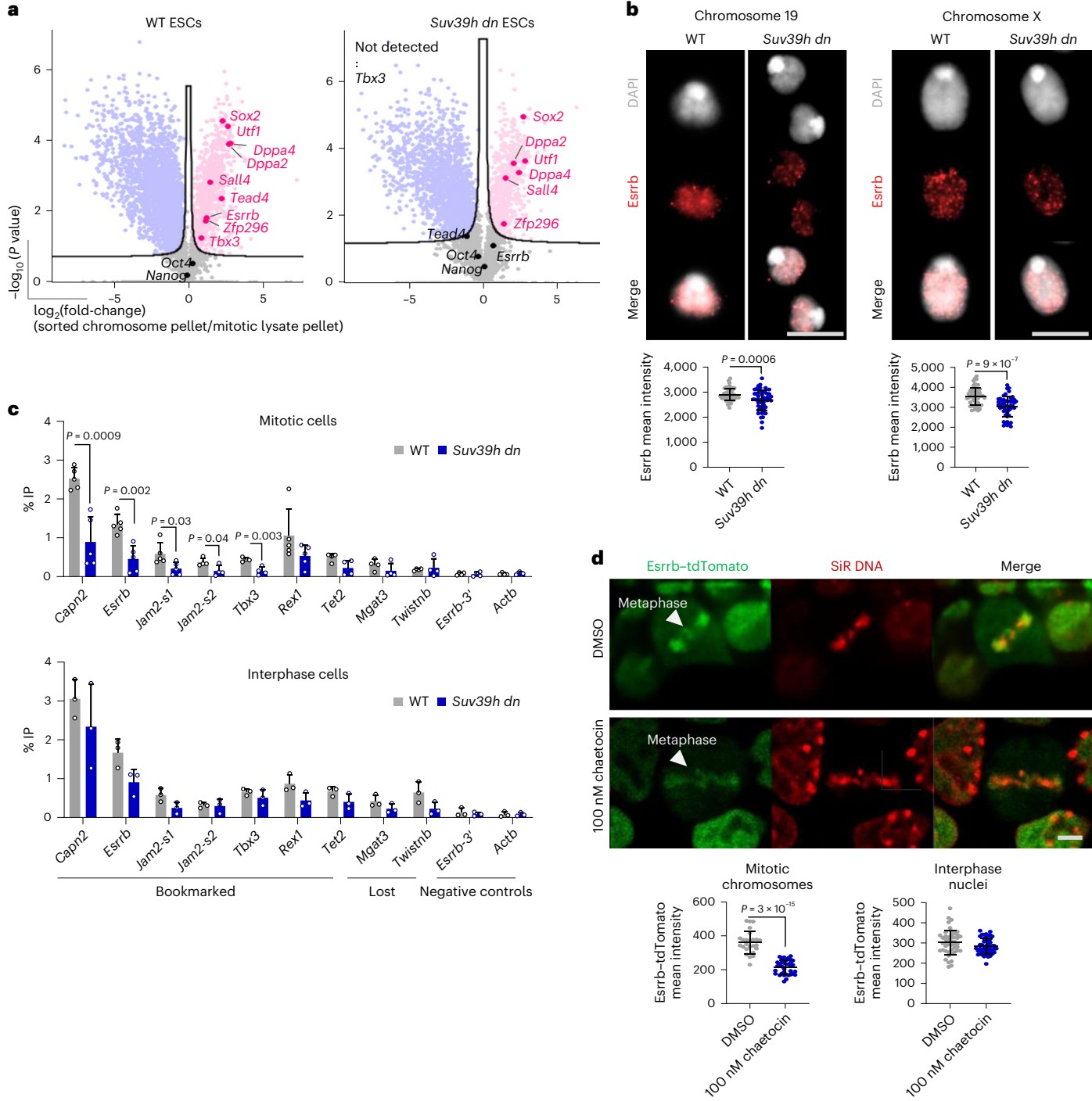

**Fig. 4 | ESCs lacking Suv39h1/Suv39h2 show an altered retention of pluripotency-associated factors on mitotic chromosomes. a**, Volcano plots as in Fig. 3d, highlighting pluripotency-associated factors that are enriched (red) or not significantly changed (black) on WT (left plot) or *Suv39h dn* (right plot) ESC mitotic chromosomes. **b**, Esrrb immunolabeling (red) of WT and *Suv39h dn* flow-sorted chromosomes19 (left panel) and X (right panel), where DAPI counterstain is shown in light gray. Scale bar, 5 μm. Esrrb mean intensities were measured across individual chromosomes; the mean ± s.d. is shown (*n* = minimum 50 chromosomes analyzed over 3 independent experiments). *P* values of statistically significant changes, measured by unpaired, two-tailed Student's *t*-tests, are indicated. **c**, Esrrb ChIP–qPCR analysis in WT versus *Suv39h dn* mitotic and asynchronous ESCs. Enrichment (immunoprecipitated as a percentage of input (%IP)) was measured at Esrrb bookmarked sites (*Capn2, Esrrb, Jam2-s1, Jam2-s2, Tbx3* and *Tet2*), Esrrb lost sites (bound only in interphase;

*Mgat3* and *Twistnb*) or control sites that do not bind Esrrb (*Esrrb-3'* and *Actb*). The mean + s.d. results are shown. For interphase cells *n* = 3 biological replicates, for mitotic cells *n* = 4 biological replicates (except *Capn2, Esrrb, Rex1* and *Jam2-s1*, where *n* = 5). **d**, Live-cell imaging of Esrrb–tdTomato mouse ESCs pretreated with DMSO (upper panel) or 100 nM of chaetocin (lower panel) cultured with SiR-DNA (red). Arrows show Esrrb localization to mitotic chromatin. Scale bar, 5 μm. Esrrb–tdTomato mean intensities on mitotic DNA (gated based on SiR-DNA signal) and in interphase nuclei were quantified for each condition; the mean ± s.d. is shown. For mitotic chromosomes: *n* = 25 (DMSO) or *n* = 35 (chaetocin) cells analyzed; for interphase nuclei: *n* = 46 cells analyzed for both DMSO and chaetocin treatments, representing 3 independent experiments. **b–d**, *P* values of statistically significant changes, measured by unpaired, two-tailed Student's *t*-tests, are indicated. Source data and precise *n* numbers are provided.

ChIP analysis of asynchronous samples showed no significant differences in Esrrb binding between these genotypes, suggesting that Esrrb binding loss is selective for (or more evident in) mitosis.

To test the impact of acute H3K9me3 depletion on mitotic retention of Esrrb, we used WT ESCs expressing an endogenous Esrrb–tdTomato fusion protein[31]. Esrrb–tdTomato ESCs were examined by live-cell imaging with and without addition of chaetocin, a mycotoxin that inhibits Suv39h1 activity[46,47] (Supplementary Videos 1 and 2).

As anticipated, Esrrb decorated metaphase chromosomes (red) in these ESCs (Fig. 4d, upper panels). However, after chaetocin treatment, Esrrb binding at metaphase chromosomes was substantially lowered (Fig. 4d, lower panel). Live-cell imaging of Esrrb–tdTomato in ESCs pretreated with chaetocin versus dimethylsulfoxide (DMSO) controls confirmed a significant decrease in signal intensity in response to chaetocin, which was evident mitosis, but not interphase (compare histograms, Fig. 4d). In contrast to Esrrb, Sox2 remained unaffected on sorted

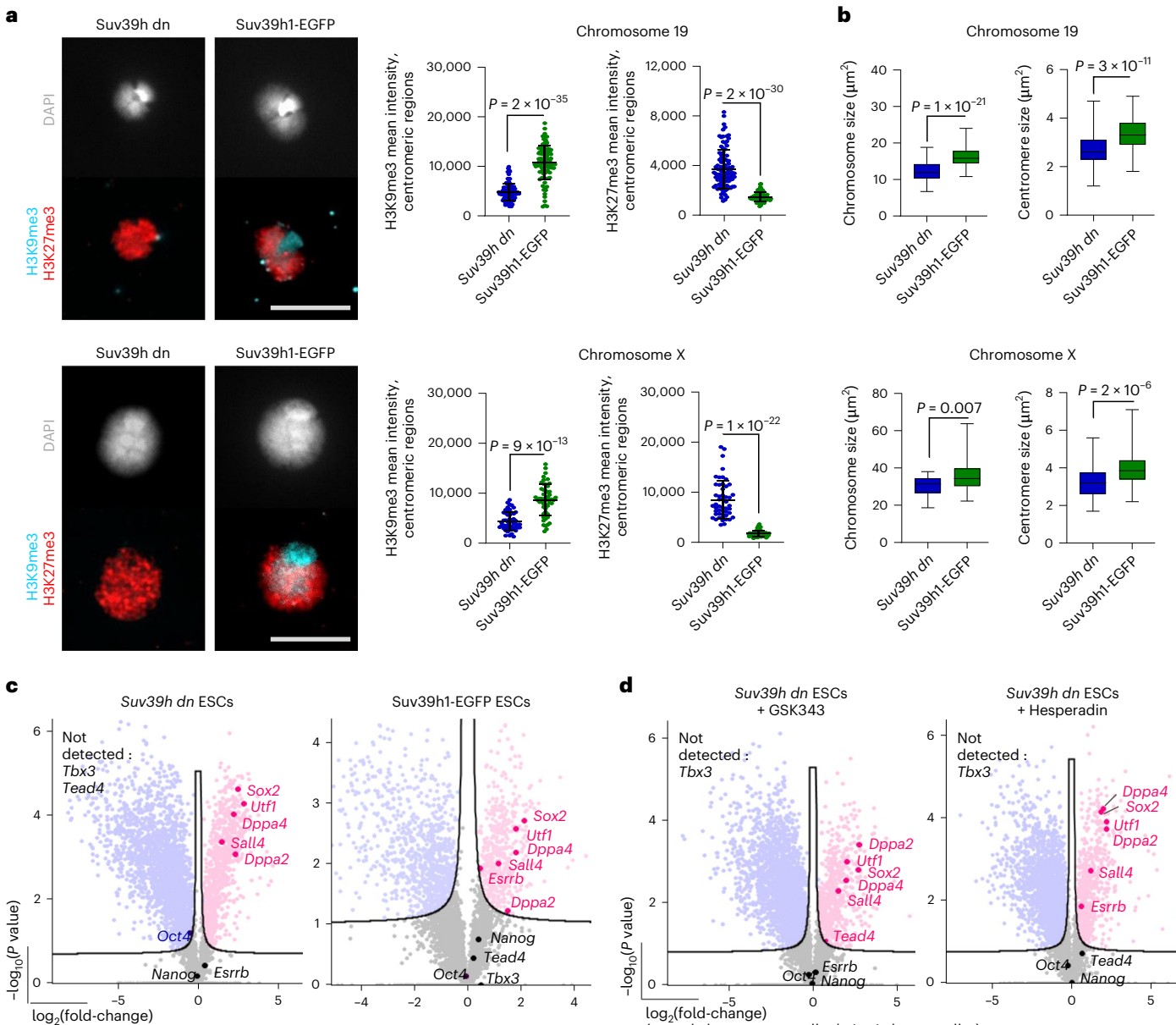

**Fig. 5 | Optimal retention of mitotic bookmarking factors such as Esrrb requires H3K9me3. a**, Representative images of coimmunolabeling of histone H3K9me3 (blue) and H3K27me3 (red) on chromosomes 19 (upper panel) and X (lower panel) isolated from *Suv39h dn* ESCs or *Suv39h dn* ESCs overexpressing Suv39h1-EGFP. The DAPI counterstain is shown in light gray. Scale bar, 5 μm. H3K9me3 and H3K27me3 mean intensities were measured at centromeric regions of chromosomes 19 and X (mean ± s.d. is shown; *n* = minimum 50 chromosomes analyzed over 3 independent experiments). *P* values of statistically significant changes, measured by unpaired, two-tailed Student's *t*-tests, are indicated. **b**, Size measurements of chromosomes 19 (upper panel) and X (lower panel) isolated from *Suv39h dn* ESCs or *Suv39h dn* ESCs overexpressing Suv39h1-EGFP. Boxplots show area measurements of individual chromosomes and centromeres for each cell line. Minimum, lower quartile, median, upper quartile and maximum values

are indicated (*n* = minimum 100 chromosome measurements for each cell line). *P* values of statistically significant changes, measured by unpaired, two-tailed Student's *t*-tests, are indicated. **a**,**b**, Source data and precise *n* numbers are provided. **c**,**d**, Volcano plots highlighting pluripotency-associated factors that are enriched (red) or not significantly changed (black) on mitotic chromosomes versus mitotic lysate pellets for *Suv39h dn* ESCs and *Suv39h dn* ESCs expressing Suv39h1-EGFP (**c**) and *Suv39h dn* ESCs treated with GSK343 or *Suv39h dn* ESCs treated with Hesperadin (**d**). Proteins were plotted as log$_2$(fold-change LFQ intensity of sorted chromosome pellets/LFQ intensity of mitotic lysate pellet and significance) (−log$_{10}$(*P*)) using Perseus software (unpaired, two-tailed Student's *t*-test, permutation-based FDR < 0.05, s0 = 0.1; *n* = 3 independent experiments each measured in technical duplicate; see Methods for details).

mitotic chromosomes after chaetocin treatment (Supplementary Fig. 4g,h), indicating that Suv39h1 drug inhibition does not induce a global dissociation of pluripotency factors. Taken together, these data show that both short- and long-term depletion of Suv39h1/Suv39h2 activity in ESCs result in reduced retention of Esrrb on mitotic chromosomes.

### Mitotic chromosome size and factor retention depend on Suv39h

Our data raise the intriguing possibility that retention of TFs through mitosis may be sensitive to H3K9me3. As this could reflect a dependency on either H3K9me3 itself or the correct marking and function of constitutive heterochromatin domains, we asked whether other H3K9me3-associated proteins were also enriched or depleted in mitotic samples (Supplementary Fig. 5a). It is interesting that HP1α (or CBx5) binding was retained on *Suv39h dn* mitotic chromosomes, as were the Swi/Snf chromatin re-modelers Smarcb1 and Atrx, a protein known to bind at heterochromatic repeat elements, including telomeres, ribosomal DNA repeats, endogenous retroviral elements and pericentric domains[48,49]. In contrast, components of the lysine-specific histone demethylase complex 1A (LSD1 or Kdm1a), which are known to interact with Esrrb in trophoblast stem cells[50], were less well retained on H3K9me3-depleted mitotic chromosomes (compare Kdm1a and Rcor1; Supplementary Fig. 5a). Our proteomic comparisons also revealed increases in the representation of certain factors, notably histone H1 variants, at mitotic *Suv39h dn* chromosomes, relative to WT controls (Supplementary Fig. 5b). This may be relevant because linker histones have been widely implicated in chromatin condensation[51–59] and an over-representation of histone H1 could contribute to the characteristically compact state of *Suv39h dn* mitotic chromosomes.

To ask whether inefficient retention of Esrrb binding during mitosis was reversed by H3K9me3 rescue, we transfected *Suv39h dn* ESCs with Suv39h1 (ref. [39]). The provision of Suv39h1 restored H3K9me3 labeling of centromeric domains, as exemplified for mitotic chromosomes 19 and X (Fig. 5a), and resulted in decreased H3K27me3 across centromeric (DAPI-intense) domains (Fig. 5a). Importantly, Suv39h1 transfection also rescued mitotic chromosome size and centromere compaction (Fig. 5b), and restored efficient Esrrb retention by *Suv39h dn* mitotic chromosomes, assessed by proteomic profiling (Fig. 5c).

### Impact of H3K9me3 loss on chromatin and TF binding in mitosis

We have shown that H3K9me3 removal results in increased levels of H3K27me3 and H3S10ph on mitotic chromosomes. As these chromatin changes might also contribute to the altered binding of factors such as Tead4 and Esrrb in *Suv39h dn* cells, we asked whether inhibition of PRC2 activity (GSK343) or aurora kinase b activity (Hesperadin)[60] (to reduce H3K27me3 or H3S10ph, respectively) impacts mitotic retention. Proteomic comparisons (Fig. 5d) showed that Esrrb retention by *Suv39h dn* mitotic chromosomes was increased by inhibiting H3S10ph, whereas Tead4 retention was selectively increased on H3K27me3 inhibition. To understand how Esrrb retention during mitosis might be impacted by the loss of H3K9me3 and altered heterochromatin structure[41], we examined the detailed distribution of Esrrb through the cell cycle, relative to heterochromatic and euchromatic chromatin features. Using the tdTomato–Esrrb ESC line[31], endogenous Esrrb clearly decorated chromosomes throughout all stages of mitosis (Supplementary Fig. 6a) and also colocalized with DAPI-stained DNA in interphase, as reported previously[31]. We show in the present study that euchromatic and heterochromatic regions of chromosomes are labeled by Esrrb (Fig. 6a and Supplementary Fig. 6a), with signal also detected at DAPI-intense, pericentromeric domains of isolated mitotic chromosomes (Fig. 6b). Treatment with the Suv39h1 inhibitor chaetocin substantially reduced Esrrb detection throughout mitosis (metaphase and telophase stages) and affected Esrrb distribution at euchromatin and heterochromatin

domains (Fig. 6c). As H3K9me3 can regulate the expression of genomic repeat elements[27], we examined the expression of euchromatin- and heterochromatin-based repeats in *Suv39h dn* mitotic ESCs. Cot-1 RNA, which predominately contains LINE-1 (long interspersed nuclear element-1) and SINE (short interspersed nuclear element) repeat elements, was used to probe euchromatin repeat expression[61] and gamma-satellite (γSat) RNA to probe heterochromatin-repeat expression[62]. We observed a marked increase in Cot-1 RNA in mitotic *Suv39h dn* ESCs compared with WT controls (pink, Fig. 6d), and more specifically LINE-1 expression, as confirmed by quantitative (q)PCR analysis (Supplementary Fig. 6b) and L1 ORF1 protein levels (Supplementary Fig. 6c). In contrast, the expression and distribution of major satellite repeats RNA (yellow, γSat; Fig. 6d) appeared similar in WT and *Suv39h dn* ESCs through mitosis.

## Discussion

To gain a broad understanding of the impact that H3K9 methylation loss has during mitosis, we examined the size and epigenetic features of chromosomes isolated from ESCs and fibroblasts that lacked Suv39h1/Suv39h2, Setdb1/Setdb2 or G9a/Glp. Mitotic chromosomes from cells devoid of H3K9me3 (Suv39h1/Suv39h2 double null) were distinct in being significantly more compact than WT equivalents, with highly condensed centromeres decorated by H3K27me3 (rather than H3K9me3). These mitotic chromosomes also showed a twofold enrichment in H3S10ph and significantly increased representation of several histone H1 variants, as determined by proteomic analysis (histones H1.1, H1.2, H1.3, H1.4 and H1.5; Supplementary Fig. 5b). Increased levels of histone H1 variants, H3S10ph and H3K27me3, have been independently shown to result in enhanced chromosome compaction in other settings[36,51–53,63–65] and we show in the present study that inhibition of H3K27me3 activity can reverse the compaction of *Suv39h dn* mitotic chromosomes. These results contribute to an extensive catalog of structural and organizational defects that have previously been described in interphase mammalian cells on H3K9me3 withdrawal[66,67]. This includes altered heterochromatin organization, extended telomere length, DNA-repair pathway activation, repeat element re-expression and genomic instability, which manifests as H3K9me3-deficient cells transit mitotic or meiotic division[13,41]. Our data, which are focused on mitotic events, reveal that H3K9me3 is also required to sustain binding of a rich cadre of proteins to mitotic chromosomes. These proteins collectively influence DNA-templated transcription, chromatin silencing, DNA repair, cellular stress and chromosome organization, processes that are likely to be particularly important during and immediately after cell division. Although we did not observe mitotic defects in our cultured ESC lines, it is worth noting that Suv39h1/Suv39h2-deficient mice show severely impaired viability and aneuploidy with increased tumor incidence, and also fail to generate mature functional sperm[13].

In the present study, we showed that H3K9me3 is required for the efficient retention of some TFs such as Esrrb on mitotic chromosomes during ESC division. Genetic ablation of Suv39h1 and Suv39h2 resulted in a reduced representation of Esrrb on native sorted mitotic chromosomes, as revealed by quantitative proteomics, as well as reduced Esrrb protein detected by cellular fluorescence-based labeling. This dependency of Esrrb was supported independently by studies of acute H3K9me3 depletion targeted by pharmacological exposure to chaetocin, and demonstrations that Esrrb representation on *Suv39h dn* mitotic chromosomes was restored by transfection of Suv39h1 and H3K9me3 rescue. These results highlight a previously unrecognized role for H3K9me3 in retaining TFs, including Esrrb, Tead4, Tbx3 and other proteins (involved in transcription, DNA repair, chromatin silencing and chromosome organization) on condensed mitotic chromosomes. The binding of Tead4, a factor involved in Hippo signaling[68], was diminished in *Suv39h dn* mitotic ESC chromosomes, as was Tbx3, a factor expressed early in the developing intracellular matrix and implicated in regulating extraembryonic endoderm[69] and the mitotic

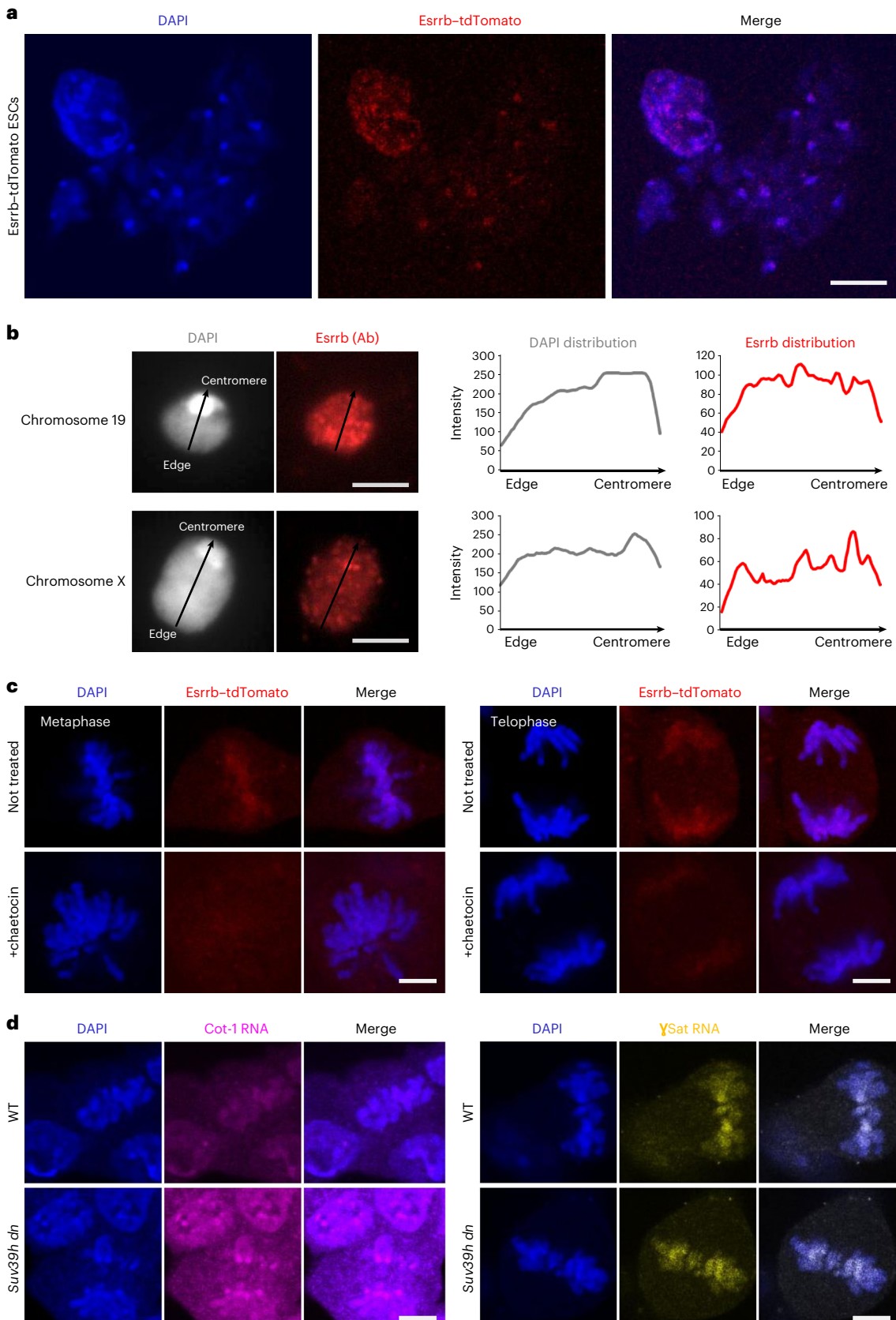

**Fig. 6 | Esrrb association with euchromatin and heterochromatin regions during mitosis. a**, Representative images of Esrrb–tdTomato ESC metaphase spreads stained with DAPI (blue). Scale bar, 10 μm. **b**, Esrrb immunolabeling (red) of flow-sorted chromosomes 19 (top panel) and X (lower panel), where DAPI counterstain is shown in light gray. Scale bar, 5 μm. Linescan analysis (profile plots) is shown of Esrrb (red) and DAPI (gray) intensities across chromosomes 19 and X (right panels). Ab, antibody. **c**, Representative fluorescence images of Esrrb–tdTomato ESCs at metaphase (left) and telophase (right) stages with and without chaetocin treatment. DAPI stain is in blue. **d**, Cot-1 (pink) and γSat repeat (yellow) RNA–FISH in WT and *Suv39h dn* ESCs, with DAPI stain in blue. Scale bar, 5 μm. All the images represent three independent experiments.

bookmarking factor Esrrb[31,32]. In contrast, the pluripotency factor Sox2, which is implicated in regulating target genes in association with either Oct4 or Esrrb[70–72], was equivalently represented on mitotic chromosomes with or without H3K9me3. Previous studies have shown that Sox2 and Esrrb normally remain chromosome bound in mitosis[30,31] and these results reveal an intriguing selective sensitivity of Esrrb to H3K9me3 depletion.

ATAC-seq comparisons of WT and *Suv39h dn* asynchronous ESCs, mitotic cells and isolated native mitotic chromosomes showed that the profiles of mutant and control samples were surprisingly similar, indicating that canonical binding sites for this protein remain largely accessible. However, ChIP analysis at selected target genes clearly showed that Esrrb binding was diminished in *Suv39h dn* mitotic chromosome samples at several 'bookmarked' target sites[31]. It is possible that changes in Esrrb binding could reflect chromatin events that are downstream of H3K9me3 loss, such as increased phosphorylation of H3S10. In this regard, we showed that increased H3S10ph correlated with reduced representation of Esrrb and the Esrrb-associated factors Rfeb1, Tfeb[50] and Tcf7l1. Furthermore, inhibition of aurora kinase B activity in *Suv39h dn* ESCs fully restored the representation of Esrrb on mitotic chromosomes, highlighting a functional overlap between H3K9me3 loss and H310Sph gain. At least two other mechanisms should also be considered. The first acknowledges that, in mitosis, extreme chromosome condensation could elicit liquid–liquid phase separation and the formation of local condensates of DNA-binding factors that may coat chromosomes. This may be impacted by Suv39h loss and the resulting changes of heterochromatin. Second, it remains formally possible that Suv39h1 and Suv39h2 have nonhistone substrates that could impact Esrrb and chromosome function[73]. For example, in B cells, Suv39h1 has been shown to methylate RAG-2 (recombination activating gene 2 protein) and is implicated in class switch recombination[74], and also methylates Dot1L and the mycobacterial protein HupB[75]. Likewise, Suv39h2 can trimethylate the histone H3K4 demethylating enzyme lysine-specific demethylase 1 (LSD1) and methylates H2AX during damage repair[76,77].

Cells lacking Suv39h1 and Suv39h2 also display changes in the repertoire of factors bound to mitotic chromosomes, providing important primary evidence that chromatin is critical for TF retention through mitosis. *Suv39h dn* fibroblasts and ESCs showed reduced representation of several TFs on H3K9me3-depleted mitotic chromosomes, in addition to proteins involved in DNA repair, nucleosome organization, chromatin silencing and chromosome organization. It is interesting to speculate that these groups of proteins might be relevant for protecting the genetic and epigenetic state of daughter cells that inherit a cargo of chromosome-associated proteins after cytokinesis, ahead of de novo gene expression beginning in G1. In this setting, both WT and *Suv39h dn* mitotic chromosomes showed an enrichment in core PRC1, PRC2 components and DNA-methylation machinery, relative to mitotic lysates. In addition, although retention of certain TFs was impaired after H3K9me3 loss, many pluripotency-associated factors, such as Dppa2, Dppa4, Mpp8 and Sox2, remained chromosome associated in Suv39h1/Suv39h2 mutants through mitosis. Dppa2 and Dppa4 are small heterodimerizing nuclear proteins that are known to regulate zygotic genome activation. In ESCs, these proteins have also been shown to be critical for maintaining the functionally primed state of a subset of bivalent genes in ESCs, protecting them from de novo DNA methylation and irreversible silencing[78,79]. M-phase phosphoprotein 8 (Mpp8 or Mphosph8) is an essential player in safeguarding ground-state pluripotency and stem-cell renewal, through interactions with a plethora of epigenetic silencing proteins, including Dnmt3a, Setdb1 and Sirt1. It is of interest that Mpp8 has also been implicated in repressing LINE-1 and L1-ORF2 expression, together with the human silencing hub (HUSH) complex[80]. Although we do not yet know the basis of mitotic factor sensitivity or resistance to H3K9me3 loss, it is possible that chromosome-bound Dppa2, Dppa4 and Mpp8 could act as safeguards, protecting the epigenetic fidelity and pluripotent state of newly established daughter cells, particularly in the context of de novo DNA methylation conveyed by chromosome-bound Dnmt3a. Our analyses also showed that ESCs that lack Suv39h1/Suv39h2 overexpress LINE-1 and L1-ORF2, consistent with previous studies[27], and we detected an increased representation of L1-OF2 protein on H3K9me3-depleted mitotic chromosomes (Supplementary Fig. 6b,c). As H3K9me3 is not strictly required for Mpp8-mediated repression of LINE-1 elements[80], it is likely that the overexpression of LINE-1 elements in mutant Suv39h1h2-null cells may stem from failures to recruit and maintain Setdb1 and the HUSH complex at appropriate targets.

Our study asked whether repressive chromatin has a role in retaining chromosome-associated factors through mitosis. This important question was addressed using an approach that enables unfixed 'native' mitotic chromosomes to be directly isolated from dividing cells[36]. By examining purified mitotic chromosomes either from cells at different stages of development or from cells that lack specific chromatin features, it was possible to compile a comprehensive and quantitative catalog of proteins retained by mitotic chromosomes in the absence of Suv39h1/Suv39h2-mediated H3K9me3. We showed that repressive chromatin is critical for maintaining the efficient binding of certain proteins to mitotic chromosomes. Therefore, in addition to the important role of H3K9me3 in regulating gene and chromosome function during interphase, H3K9me3 is important for mitotic chromosome structure and the efficient retention of a cohort of TFs during mitosis.

## Online content

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

## Methods

### Cells

Mouse ESCs used were WT (WT26), *Suv39h dn* (DN57)[26], *Suv39h1-EGFP* (*Suv39h dn* ESCs overexpressing full-length Suv39h1)[39] and Esrrb–tdTomato[31] (gift from N. Festuccia). ESC lines were grown on 0.1% gelatin (Sigma-Aldrich)-coated dishes in KnockOut Dulbecco's modified Eagle's medium (KO-DMEM) supplemented with 15% fetal bovine serum (FBS), L-glutamine, nonessential amino acids, 2-mercaptoethanol, antibiotics and leukemia inhibitory factor (made in-house). MEFs used in the present study were WT (W8), *Suv39h dn* (D5)[13], WT (Eset25 control), *Suv39h1/2⁻/⁻* (B1), *Setdb1/2⁻/⁻* (A4 + 4-OHT) and *G9a/Glp⁻/⁻* (H7)[41]. MEFs were maintained in DMEM supplemented with 10% FBS, L-glutamine, antibiotics, 2-mercaptoethanol, nonessential amino acids and sodium pyruvate. For A4 cells, Setdb1 deletion was induced by plating cells in 2 μM 4-OHT for 2 d, followed by 2 days of recovery in complete medium without tamoxifen. All cell lines used in the present study are male and equivalent passage between WT and mutants.

### Drug treatments

After passaging, cells were treated for 24 h with 1 μM GSK343 (Ezh2 inhibitor, Sigma-Aldrich), 100 nM 5-Aza (DNA methylation inhibitor, Sigma-Aldrich), 100 nM chaetocin (Suv39h1 inhibitor, Enzo-LifeScience) or 100 nM Hesperadin (Aurkb inhibitor, Sigma-Aldrich), from stock solutions prepared in DMSO.

### Mitotic chromosome preparation and flow sorting

Mitotic chromosomes were prepared using a polyamine-based method[36,81]. Mitotic arrest, chromosome preparation and flow sorting were performed as previously described[36]. Cells were arrested using 0.1 μg ml⁻¹ of demecolcine (Sigma-Aldrich) for 6 h at 37 °C. Mitotic cells were collected by mitotic shake off, pelleted at 289*g* for 5 minutes (min) and resuspended in 10 ml of hypotonic solution (75 mM KCl, 10 mM MgSO₄, 0.5 mM spermidine trihydrochloride and 0.2 mM spermine tetrahydrochloride, adjusted to pH 8) for 20 min at room temperature (RT). Swollen cells were centrifuged at 300*g* for 5 min at RT, resuspended in 3 ml of ice-cold polyamine buffer (15 mM Tris-HCl, 2 mM EDTA, 0.5 mM (ethylenebis(oxonitrilo))tetra-acetate (EGTA), 80 mM KCl, 3 mM dithiothreitol, 0.25% Triton X-100, 0.2 mM spermine tetrahydrochloride and 0.5 mM spermidine trihydrochloride, pH 7.7) and incubated for 15 min on ice. Mitotic chromosomes were released by vortexing (30 s, maximum speed) and syringing through a 22.5-gauge needle. Chromosome suspensions were centrifuged for 2 min at 200*g* and RT. The supernatant containing mitotic chromosomes was filtered through a 20-μm mesh CellTrics filter (Sysmex). Mitotic chromosomes were stained with 5 μg ml⁻¹ of Hoechst 33258 (Sigma-Aldrich), 50 μg ml⁻¹ of chromomycin A3 (Sigma-Aldrich) and 10 mM MgSO₄, overnight at 4 °C. Sodium citrate (10 mM final) and sodium sulfite (25 mM final) were added to chromosome suspensions 1 h before FACS sorting. Chromosomes were analyzed and sorted using a Becton Dickinson Influx equipped with spatially separated lasers using BD FACS software. As previously described[36], Hoechst 33258 was excited using a 355-nm air-cooled laser (Spectra Physics Vanguard) with a power output of 350 mW, and fluorescence was collected using a 400-nm long-pass filter in combination with a 500-nm short-pass filter. Chromomycin A3 was excited using a water-cooled, 460-nm laser (Coherent Genesis) with a power output of 500 mW. Chromomycin A3 fluorescence was collected using a 500-nm long-pass filter in combination with a 600-nm short-pass filter. As previously described[36], forward scatter was measured using a Coherent Sapphire 488-nm laser with a power output of 200 mW and this was used as the trigger signal for data collection. Chromosomes were sorted using a 70-μm nozzle tip, a drop drive frequency of ~96 kHz and a sheath pressure of 65 p.s.i (448 kPa).

### Immunofluorescence on flow-sorted chromosomes

Flow-sorted chromosomes (10⁵) were cytospun (Cytospin3, Shandon) on to poly(L-lysine) slides at 1,200 r.p.m. for 10 min, as previously described[36]. Samples were incubated with blocking buffer (5% normal goat serum in 10 mM Hepes, 2 mM MgCl₂, 100 mM KCl and 5 mM EGTA) for 30 min at RT, and incubated overnight at 4 °C with primary antibodies to Cenpa (catalog no. 2048S, Cell Signaling), H3K9me3 (catalog no. 07-523 or 07-442, Millipore), H3K27me3 (catalog no. ab6002, Abcam or catalog no. 07-449, Millipore), Esrrb (catalog no. PP-H6705-00, Perseus Proteomics) and Sox2 (catalog no. ab97959, Abcam). All antibodies were diluted 1:200 in blocking buffer. Chromosomes were incubated with appropriate secondary antibodies (anti-mouse Alexa 488 (catalog no. A11001, Invitrogen), anti-rabbit Alexa 488 (catalog no. A11008, Invitrogen) or anti-mouse Alexa 568 (catalog no. A11031, Invitrogen)) for 1 h at RT. All secondary antibodies were diluted 1:400 in blocking buffer. Stained chromosomes were mounted in DAPI-containing Vectashield (Vector Laboratories). Wide-field epifluorescence microscopy was performed on an Olympus IX70 inverted microscope using a UPlanApo ×100/1.35 oil objective lens and Micro-Manager software.

### Immunofluorescence on ESCs

ESCs were cultured on gelatin-coated glass coverslips, fixed with 1% formaldehyde 10 min at RT and crosslinked with 2 mM disuccinimidyl glutarate (DSG, Sigma-Aldrich) for 50 min at RT, as previously described[32]. Cells were permeabilized with 0.1% Triton X-100 for 15 min at RT and blocked with 1% bovine serum albumin and 5% goat serum. Cells were incubated with primary antibodies Esrrb or H3S10ph (catalog no. ab5176, Abcam), 1:200, at 4 °C overnight. After three phosphate-buffered saline (PBS) washes, cells were labeled with appropriate secondary antibodies and mounted in DAPI-containing Vectashield. Images were acquired with an Olympus IX70 inverted microscope using a UPlanApo ×100/1.35 oil objective lens and Micro-Manager software.

### Chromosome size and histone modification quantification

Chromosome images acquired on the Olympus IX70 wide-field microscope were analyzed in Fiji[82]. Chromosome and centromere size measurements were performed as previously described[36] by estimating whole chromosome (total DAPI) and centromere (DAPI high) areas. H3K27me3 and H3K9me3 maximum intensity projections of *z*-planes were quantified for both centromere and whole chromosome areas of each individual chromosome.

### Metaphase spreads

Metaphase spreads were prepared using the following method, as previously described[36]. Metaphase-arrested cells were resuspended in hypotonic solution (40 mM KCl, 0.5 mM EDTA, 20 mM Hepes, pH 7.4) for 25 min at 37 °C. Nuclei were pelleted (8 min at 500*g*) and supernatant was removed, leaving a small drop for resuspension. Next, a 3:1 mixture of MeOH and glacial acetic acid fixative was added to the top of the tube and tubes were stored at −20 °C. The next day nuclei were pelleted (8 min at 500*g*) and washed in fresh 3:1 MeOH:glacial acetic acid 3×. To prepare metaphase spreads, nuclei were pelleted and resuspended in a small volume of fixative (to a pale-gray solution). A 20-μl drop of 45% acetic acid in water was pipetted on to a glass Twinfrost microscope slide and 23 μl of spread mixture was dropped on to it, tilting the slide to spread the nuclei. Slides were air dried at RT. Mouse chromosome X and 19 painting probes (Metasystems Probes) were used together with mouse γSat probes (DNA a gift from N. Dillon) labeled with FluoroRed (Amersham Life Science) by nick translation, to detect chromosome X or 19 with pericentromeric DNA. Metaphase chromosome painting was performed according to the Metasystems Probes protocol and samples were mounted in Vectashield containing DAPI. A Leica SP5 II confocal microscope was used for imaging using LAS-AF software.

## ATAC-seq

ATAC-seq[83] was performed in duplicate on asynchronous and mitotic WT and *Suv39h dn* ESCs and on purified mitotic chromosomes, with the method details as previously described[36]. Nuclei were obtained from 50,000 asynchronous ESCs according to the Omni-ATAC-seq protocol[84] by resuspending in 50 μl of ATAC-resuspension buffer (10 mM Tris-HCl, pH 7.4, 10 mM NaCl, 3 mM MgCl$_2$, 0.1% Igepal CA-630, 0.1% Tween-20 and 0.01% digitonin) and incubating on ice for 3 min. After adding 1 ml of ATAC-resuspension buffer without Igepal and digitonin, nuclei were pelleted at 500$g$ (10 min at 4 °C). To perform ATAC-seq, nuclei, mitotic cells (50,000) or purified chromosomes (2 × 10$^6$ pelleted at 10,000$g$ for 10 min at 4 °C) were resuspended in 50 μl of transposase mixture (25 μl of Illumina TD buffer, 22.5 μl of H$_2$O, and 2.5 μl of Illumina TDE1 transposase) and incubated at 37 °C for 30 min, shaking at 1,000 r.p.m. DNA was purified and amplified with seven PCR cycles (NEBNext High Fidelity master mix; primers in Supplementary Table 1)[83]. Libraries were purified twice with Ampure XP beads (Beckman Coulter), including size selection with 0.5× beads. Libraries were assessed by Qubit, Bioanalyzer and KAPA Library Quantification (Roche) before Illumina NextSeq500 sequencing (75 bp, paired end).

Data were processed using RTA (v.2.11.3, default settings) and reads were demultiplexed with bcl2fastq2 (v.2.20.0; allowing 0 mismatches). Reads were trimmed with Trim Galore! (trim_galore_v0.4.4; --trim-n --paired; www.bioinformatics.babraham.ac.uk/projects/trim_galore) and aligned to University of California Santa Cruz (UCSC) mm10 using bowtie2 (bowtie2/2.2.9; -p8 -t -very-sensitive -X 2000)[85]. Aligned bam files were sorted with Samtools (v.1.2)[86] and technical replicate sequencing runs were merged. Duplicate reads were marked with Picardtools MarkDuplicates (v.1.90; http://broadinstitute.github.io/picard). The bamQC function from R/Bioconductor package ATAC-seqQC[87] was used to keep properly mapped paired-end reads, and remove duplicates and mitochondrial alignments. Chromosome-wide accessibility profiles and accessibility trend plots were generated from the resulting bam files in Seqmonk (v.1.48.0; www.bioinformatics.babraham.ac.uk/projects/seqmonk) as previously described[36]. Accordingly, chromosome-wide accessibility profiles show transposase insertion frequency (5′-read ends, offset +4 bp/−5 bp) in each 25-kb window, relative to the genome-wide average. Accessibility trend plots show average relative distribution of insertion sites (extended ±25 bp) across 2-kb windows centered on Esrrb peak summits (taken from ref. [32] and converted to mm10 using the UCSC LiftOver tool). Bam files were further processed with deepTools2 (ref.[88]) by using the alignmentSieve function to shift and extract nucleosome-free fragments (<100 bp). Peaks were called from nucleosome-free fragments with model-based analysis for ChIP-seq 2 (MACS2: -f BAMPE)[89]. Peaks overlapping ENCODE blacklist v.2 regions[90] were removed. For downstream analysis, we separately defined consensus peaks for cells and purified chromosomes. For each, we defined nonredundant peaks using the reduce function from R/Bioconductor package GenomicRanges[91] and only kept peaks appearing in at least two samples. Peak-based read counts were obtained with the featureCounts function from R/Bioconductor package Rsubread[92,93] and normalized using the calcNormFactors function (method = 'TMM') in R/Bioconductor package edgeR (v.3.28.1)[94,95]. Differential accessibility analysis was performed with R/Bioconductor package limma (v.3.42.2)[96], after applying the voom function[97] to estimate the mean-variance relationship within the count-based datasets.

## Proteomics

Proteomics was performed on mitotic lysate pellets and mitotic chromosomes from WT and *Suv39h dn* MEFs and ESCs, *Suv39h dn* ESCs treated with GSK343 or Hesperadin and *Suv39h1-EGFP* ESCs, using a method previously described[36]. Samples were digested with trypsin by an in-Stage Tip digestion protocol[98] using commercially available iST Kits (Preomics) according to the manufacturer's recommendations.

Protein digests were analyzed by LC-MS/MS using a data-dependent acquisition method with a 50-cm EasySpray column at a flow rate of 250 nl min$^{-1}$ coupled to a Q-Exactive HF-X mass spectrometer, as described previously[36]. Experiments were carried out in biological triplicate and digests were analyzed by LC-MS/MS as technical duplicates.

Raw data were analyzed using MaxQuant[99,100] with the in-built LFQ algorithm[101]. Statistical analysis and data visualization were performed using Perseus software[102]. Analysis details are provided as previously described[36], with minor modifications. After processing raw proteomic data in MaxQuant, the proteinGroups.txt file was imported into Perseus (v.1.6.2.2 or v.1.6.7.0) with the respective LFQ intensities as the main columns. We filtered on categorical columns to remove reverse decoy hits, potential contaminants and protein groups that were 'only identified by site'. GO annotations for taxonomy *Mus musculus* (mainAnnot.mus_musculus.txt) were downloaded from http://annotations.perseus-framework.org. GO annotations for molecular function and cellular compartment were imported by annotating columns based on majority protein IDs. Groups of technical replicate injections and biological replicates of the two conditions ('mitotic lysate pellet' and 'sorted chromosome pellet') were defined in categorical annotation rows. Missing values were replaced with 'NaN' (Quality→Convert to NaN), technical duplicates were averaged (annot. rows→average groups→mean value; min. one valid value) and data were log(transformed) (Basic→Transform→log$_2(x)$). Principal component analysis of proteomic datasets was performed after the filtering steps. Volcano plots were generated based on LFQ intensities with the following settings: test, Student's $t$-test; side, both; number of randomizations, 250; preserve grouping in randomizations, <None>; permutation-based FDR, 0.05; and s0, 0.1. Hierarchical clustering analysis was carried out after filtering rows based on a minimum of two valid values in at least one group, $z$-scoring of values in rows and a two-sample Student's $t$-test of the conditions using the following settings: test, Student's $t$-test; s0, 0; side, both; and permutation-based FDR, 0.05. After filtering rows retaining Student's $t$-test significant hits only, hierarchical clustering analysis was generated with the following settings for both the rows tree and the columns tree: distance, Euclidean; linkage, average; constraint, none; preprocess with $k$-means selected (number of clusters, 300; maximal number of iterations, 10; number of restarts, 1).

Majority protein IDs were extracted from the list of enriched hits from the volcano plots (listed in Supplementary Data 1 and 2) and were used to generate Venn diagrams and in GO analysis (http://geneontology.org; Fisher's exact test with FDR correction).

## Comparison of transcript abundance with proteomics

Raw poly(A) RNA-seq data from WT and *Suv39h dn* ESCs were obtained from the National Center for Biotechnology Information (NCBI) Sequence Read Archive (SRA; accession no. GSE57092)[27]. Reads were aligned to the UCSC mm10 genome using the STAR aligner (v.2.7.7a)[103] based on the Ensembl genome annotation (v.2.7.7)[104]. Gene-based read counts were obtained with STAR and normalized by calculating transcripts per million. Differential expression analysis was performed using DESeq2 (v.1.30.1)[105] to generate log$_2$(fold-changes). Gene symbols were obtained for transcripts using Ensembl BioMart (https://www.ensembl.org/biomart/martview) and used to match transcriptomics and proteomics data. Ambiguous gene symbols were removed from the proteomics data. In total, 5,664 genes were mapped between proteomics and transcriptomics datasets for correlation analyses.

## Live-cell imaging

Esrrb–tdTomato ESCs were plated in 0.1% gelatin-coated, 8-well Ibidi μ-Slides. On the next day, cells were treated with 100 nM chaetocin or DMSO in fresh ESC medium for 24 h. Cells were switched to fully supplemented phenol red-free medium, buffered with 20 mM Hepes, 1 h before imaging. SiR-DNA (1 μM, SpiroChrome) was added 30 min

before imaging. Time-lapse images were acquired on an Olympus IX70 inverted wide-field microscope using a UPlanApo ×100/1.35 oil objective lens and an environmental chamber kept at 37 °C with a 5% $CO_2$ supply. The z-stacks were collected every 120 s with a step size of 2 µm. Images were deconvolved with Huygens Professional (v.19.10, Scientific Volume Imaging, http://svi.nl), using the classic maximum likelihood estimation algorithm. Esrrb–tdTomato fluorescence signal was measured in Fiji[82] on maximum intensity projections of z-planes for both interphase nuclei and metaphase chromosomes.

### Probe labeling
Mouse Cot-1 (Invitrogen) or γSat DNA (a gift from N. Dillon), 1 µg, was labeled with Cy3-dUTP or Cy5-dUTP (Cytiva or Sigma-Aldrich) using a Prime-It Flour Fluorescence labeling kit (v.B, Agilent Technologies) according to the manufacturer's instructions. Briefly, a DNA template was annealed with random 9-mer primers at 95 °C for 5 min, then the RNA extended using exonuclease-free Klenow polymerase for 30 min at 37 °C. The reaction was stopped and the probe stored at 4 °C.

### RNA–FISH
Cells were grown on coverslips, fixed using 2.6% formaldehyde in PBS for 10 min, then permeabilized using 0.4% Triton X-100 in PBS for 5 min on ice. Cells were washed with PBS and dehydrated using an ethanol series (3 min in 70%, 80%, 95% and 100% EtOH). RNA probe, 2.5 µl, in 12 µl of hybridization buffer (1 ml 50% Denhardt's solution, 200 µl of saline–sodium citrate (SSC) containing 20% dextran sulfate, 800 µl of formamide) was used for each coverslip of 22 × 22 mm². Probe was denatured at 75 °C for 8 min, then placed on ice for 2 min. Cells were inverted on to the probe on a slide, sealed with rubber cement and placed in a humid chamber at 37 °C overnight. After hybridization, the rubber cement was removed and cells washed 3× for 3 min with 2× SSC containing 50% formamide at 42 °C followed by 3× 3-min washes with 2× SSC at 42 °C, and briefly rinsed in water before mounting in DAPI-containing Vectashield. Images were acquired with a Leica SP5 II confocal microscope using LAS-AF software.

### ChIP
WT and *Suv39h dn* asynchronous and metaphase-arrested ESCs ($5 × 10^6$) were crosslinked in 1 ml of 2 mM DSG (Sigma-Aldrich) for 50 min at RT with occasional shaking. After PBS washes, cells were fixed in 1 ml of PBS, 1% methanol-free formaldehyde (Thermo Fisher Scientific) for 10 min at RT as previously described for Esrrb ChIP[31,32]. Cells were quenched with 1× glycine (Active Motif) and washed once with cold PBS containing protease inhibitor cocktail (Active Motif). Cell pellets were resuspended in 50 µl of lysis buffer containing 1% sodium dodecylsulfate, 10 mM EDTA and 50 mM Tris, pH 8.1, supplemented with 1 µl of protease inhibitor cocktail and incubated on ice for 15 min. Samples were diluted by the addition of 320 µl of ice-cold shearing buffer (Active Motif) as described in ref. [106], and sonicated in 1.5-ml tubes (Diagenode) using a Bioruptor Pico (Diagenode) for 7 cycles of 30 s on:30 s off at maximum power, in circulating ice-cold water. Sheared chromatin samples were centrifuged for 20 min at 18,000g and 4 °C, and supernatants were carefully transferred to fresh microcentrifuge tubes (Active Motif). Then, 50 µl of this sheared chromatin was used to assess DNA concentration and sonication efficiency (typically DNA between 200 and 800 bp) using ChIP-IT Express kit (Active motif) and ChIP clean & concentrator kit (Zymo research).

Immunoprecipitation was performed using the ChIP-IT Express kit according to the manufacturer's recommendations and 1 µg of Esrrb mouse monoclonal antibody (catalog no. PP-H6705-00, Perseus Proteomics). Input and immunoprecipitated DNA fractions were purified using ChIP clean & concentrator kit. Real-time qPCR (BioRad, CFX96 system with CFX manager software) was performed in technical triplicate for each biological replicate, using SYBR Green PCR Master Mix (QIAGEN) and the primers listed in Supplementary Table 2.

### Reporting summary
Further information on research design is available in the Nature Portfolio Reporting Summary linked to this article.

## Data availability
ATAC-seq data generated in the present study have been deposited at the GEO SRA under accession no. GSE195767. The MS proteomics data have been deposited to the ProteomeXchange Consortium via the PRIDE[107] partner repository with the dataset identifier PXD039521. Raw poly(A) RNA-seq data from *WT* and *Suv39h dn* ESCs, corresponding to accession no. GSE57092, were obtained from the National Center for Biotechnology Information SRA. UCSC mm10 genome and annotation files were retrieved from Illumina iGenome (http://igenomes.illumina.com.s3-website-us-east-1.amazonaws.com/Mus_musculus/UCSC/mm10/Mus_musculus_UCSC_mm10.tar.gz). The ENCODE mm10-blacklist.v2 used to filter peaks is available at https://github.com/Boyle-Lab/Blacklist. GO annotations for use in Perseus were downloaded from http://annotations.perseus-framework.org (mainAnnot.mus_musculus.txt). All other relevant data supporting the key findings of the present study are available within the article and its supplementary files or from the corresponding author upon reasonable request. Proteomics data are provided in Supplementary Data 1 and 2. Source data are provided with this paper.

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

## Acknowledgements

We thank the London Institute of Medical Sciences (LMS)/National Institute for Health Research Imperial Biomedical Research Centre Flow Cytometry Facility, as well as the LMS microscopy facility and the LMS Genomics and LMS Bioinformatics facilities for support. We thank N. Festuccia and P. Navarro (Institut Pasteur, Paris) for providing the Esrrb–tdTomato ESC line. We thank N. Dillon (MRC LMS) for the mouse γSat DNA plasmids. Research in the laboratory of T. Jenuwein is supported by the Max Planck Society and by additional funds from the German Research Foundation within the CRC992 consortium 'MEDEP'. The present study was funded by core support from the Medical Research Council UK to the LMS (nos. MC_U120027516, MC_UP_1605/12 and MC_UP_1605/11 to A.G.F and M.M.). D.D. was supported by a Wellcome Trust Institutional Strategic Support Fund Springboard award (no. WCMA_PSN102). A.D. is supported by a Kay Kendall Leukaemia Fund Junior Research Fellowship (no. KKL1334).

## Author contributions

A.G.F. and D.D. designed the study, wrote the manuscript and supervised the work. D.D. performed most of the experiments, conducted data analysis and designed the figures. A.D. conducted experiments, analyzed data and contributed to writing the manuscript. S.C., K.B. and N.V. conducted experiments. H.K. and A.M. conducted MS and helped with proteomic analysis. B.P. conducted all of the chromosome flow sorting. C.W. helped with live-cell imaging experiments and analysis. Y.W. and M.E.F. performed bioinformatic analysis. T.J., T.M. and M.M. contributed to the design of the study and provided scientific advice and support with experiments.

## Competing interests

The authors declare no competing interests.

## Additional information

**Correspondence and requests for materials** should be addressed to Dounia Djeghloul or Amanda G. Fisher.

**Peer review information** *Nature Structural & Molecular Biology* thanks Michiel Vermeulen and the other, anonymous, reviewer(s) for their contribution to the peer review of this work. Carolina Perdigoto, Beth Moorefield and Dimitris Typas were the primary editors on this article and managed its editorial process and peer review in collaboration with the rest of the editorial team. Peer reviewer reports are available.

# Reporting Summary

## Statistics

For all statistical analyses, confirm that the following items are present in the figure legend, table legend, main text, or Methods section.

| n/a | Confirmed | |
|---|---|---|
| ☐ | ☒ | The exact sample size (*n*) for each experimental group/condition, given as a discrete number and unit of measurement |
| ☐ | ☒ | A statement on whether measurements were taken from distinct samples or whether the same sample was measured repeatedly |
| ☐ | ☒ | The statistical test(s) used AND whether they are one- or two-sided *Only common tests should be described solely by name; describe more complex techniques in the Methods section.* |
| ☐ | ☒ | A description of all covariates tested |
| ☒ | ☐ | A description of any assumptions or corrections, such as tests of normality and adjustment for multiple comparisons |
| ☐ | ☒ | A full description of the statistical parameters including central tendency (e.g. means) or other basic estimates (e.g. regression coefficient) AND variation (e.g. standard deviation) or associated estimates of uncertainty (e.g. confidence intervals) |
| ☐ | ☒ | For null hypothesis testing, the test statistic (e.g. *F*, *t*, *r*) with confidence intervals, effect sizes, degrees of freedom and *P* value noted *Give P values as exact values whenever suitable.* |
| ☒ | ☐ | For Bayesian analysis, information on the choice of priors and Markov chain Monte Carlo settings |
| ☒ | ☐ | For hierarchical and complex designs, identification of the appropriate level for tests and full reporting of outcomes |
| ☒ | ☐ | Estimates of effect sizes (e.g. Cohen's *d*, Pearson's *r*), indicating how they were calculated |

*Our web collection on statistics for biologists contains articles on many of the points above.*

## Software and code

Policy information about availability of computer code

| Data collection | BD FACS software (v1.2.0.142) and BD DIVA (v8.0.1) were use to collect FACS data.<br>Micro-Manager (v1.4.22, IX70 microscope), LAS-AF (2.7.3.9723, SP5 II microscope), cellSens Dimension (v2.3, IX83 microscope) were used to collect imaging data.<br>Image Studio software (v4.0.21) was used for fluorescent western blot imaging.<br>Quantitative real-time PCR data was collected using Bio-Rad CFX Manager software (v3.1)<br>Sequencing data were collected on an Illumina NextSeq500. |
|---|---|
| Data analysis | ImageJ/Fiji (version 1.52p) was used for image analysis including chromosome and centromere size measurements and mitotic duration analysis.<br>Image deconvolution was performed with Huygens Professional (v19.10, Scientific Volume Imaging, http://svi.nl), using the CMLE algorithm.<br>Proteomics data were analyzed using the Label-Free Quantification algorithm in the MaxQuant software platform (v1.6.2.3). The Perseus software (v1.6.2.2 and v1.6.7.0) was used for both statistical analysis and data visualization of the proteomics results.<br>ATAC-seq data were processed using RTA (v2.11.3), reads were demultiplexed with bcl2fastq2 (v2.20.0), trimmed with Trim Galore! (trim_galore_v0.4.4; --trim-n --paired; www.bioinformatics.babraham.ac.uk/projects/trim_galore) and aligned with bowtie2 (bowtie2/2.2.9; -p8 -t -very-sensitive -X 2000). Bam files were processed with Samtools (v1.2). Duplicates were marked with Picardtools MarkDuplicates (v1.90). The bamQC function from R/Bioconductor package ATACseqQC (v1.10.4) was used to keep properly mapped paired-end reads, and remove duplicates and mitochondrial alignments. Chromosome-wide accessibility profiles and accessibility trend plots were generated from the resulting bam files in Seqmonk (v1.48.0, www.bioinformatics.babraham.ac.uk/projects/seqmonk). Nucleosome-free fragments were extracted using the alignmentSieve function from deepTools2.0. Peaks were called from nucleosome-free fragments with MACS2 (-f BAMPE). Consensus peak lists were defined using the reduce function from R/Bioconductor package GenomicRanges (v1.38.0). Peak-based read counts were obtained with the featureCounts function from R/Bioconductor package Rsubread (v2.0.1) and normalised using the calcNormFactors function (method = "TMM") in R/Bioconductor package edgeR (v3.28.1). Differential accessibility analysis was performed with R/Bioconductor package limma (v3.42.2), after applying the voom function.<br>RNA-seq data were anlaysed using the STAR aligner (v2.7.7a), gene-based read counts were obtained with STAR and normalised by calculating |

Transcripts Per Million (TPM). Differential expression analysis was performed using DESeq2 (v1.30.1).
The UCSC LiftOver tool was used to convert mm9 to mm10 coordinates (https://genome.ucsc.edu/cgi-bin/hgLiftOver).
Gene Ontology analysis was performed at http://geneontology.org/.
Gene symbols were obtained for transcripts using Ensembl BioMart (https://www.ensembl.org/biomart/martview).

For manuscripts utilizing custom algorithms or software that are central to the research but not yet described in published literature, software must be made available to editors and reviewers. We strongly encourage code deposition in a community repository (e.g. GitHub). See the Nature Portfolio guidelines for submitting code & software for further information.

## Data

Policy information about availability of data

All manuscripts must include a data availability statement. This statement should provide the following information, where applicable:

- Accession codes, unique identifiers, or web links for publicly available datasets
- A description of any restrictions on data availability
- For clinical datasets or third party data, please ensure that the statement adheres to our policy

ATAC-seq data generated in this study have been deposited at Gene Expression Omnibus under accession number GSE195767.
The mass spectrometry proteomics data have been deposited to the ProteomeXchange Consortium via the PRIDE107 partner repository with the dataset identifier PXD039521.
Raw poly-A RNA-sequencing data from WT and Suv39h dn ESCs, corresponding to GSE57092, were obtained from the NCBI Sequence Read Archive (SRA).
UCSC mm10 genome and annotation files were retrieved from Illumina iGenome (http://igenomes.illumina.com.s3-website-us-east-1.amazonaws.com/Mus_musculus/UCSC/mm10/Mus_musculus_UCSC_mm10.tar.gz).
The ENCODE mm10-blacklist.v2 used to filter peaks is available at https://github.com/Boyle-Lab/Blacklist.
Gene Ontology annotations for use in Persues were downloaded from http://annotations.perseus-framework.org (mainAnnot.mus_musculus.txt).
All other relevant data supporting the key findings of this study are available within the article and its supplementary files or from the corresponding author upon reasonable request. Proteomics data are provided in Supplementary Data 1 and 2. Source Data are provided for Figures 1b,c,e-h, 2a-c,e,f, 4b-d, 5a,b (Source Data file) and Supplemental Figures S1e, S2a-d,f,g, S4c-e,g,h, S5b, S6b,c (Supplementary Data 3).

# Field-specific reporting

Please select the one below that is the best fit for your research. If you are not sure, read the appropriate sections before making your selection.

☒ Life sciences    ☐ Behavioural & social sciences    ☐ Ecological, evolutionary & environmental sciences

For a reference copy of the document with all sections, see nature.com/documents/nr-reporting-summary-flat.pdf

# Life sciences study design

All studies must disclose on these points even when the disclosure is negative.

| | |
|---|---|
| Sample size | We used minimum n=3 for the FACS and proteomics since analysis and visualization of the proteomics data with the aid of either volcano plots or a heatmap and hierarchical clustering requires a minimum of three biological replicates for the t-test. The data were highly consistent between replicates such that n=3 was sufficient to define a large number of significant differences. ATAC-seq was performed in duplicate as the minimum required for statistical comparisons whilst keeping sequencing costs down. For experiments involving imaging cells or chromosomes, analysis is from three independent experiments to ensure reproducibility, with the total number of cells or chromosomes specified for each graph. Expression and ChIP qPCR experiments were performed in biological triplicate to ensure reproducibility and allow statistical comparisons. |
| Data exclusions | There was no exclusion/inclusion of samples in the analysis. All replicate attempts were successful. |
| Replication | All FACS analysis and imaging experiments were performed in a minimum of three biological replicates for each cell line. Proteomics analysis was performed in three biological replicates, with each analyzed in technical duplicate. Expression and ChIP qPCR measurements were performed in technical triplicate for each of at least three independent biological replicates. Western blots were repeated for three biological replicates. All replicate attempts were successful. |
| Randomization | Randomization was not relevant to this study as there was no assignment of samples to different experimental groups. |
| Blinding | Blinding was not relevant since there was no assignment of samples to different experimental groups. |

# Reporting for specific materials, systems and methods

We require information from authors about some types of materials, experimental systems and methods used in many studies. Here, indicate whether each material, system or method listed is relevant to your study. If you are not sure if a list item applies to your research, read the appropriate section before selecting a response.

## Materials & experimental systems

| n/a | Involved in the study |
|---|---|
| ☐ | ☒ Antibodies |
| ☐ | ☒ Eukaryotic cell lines |
| ☒ | ☐ Palaeontology and archaeology |
| ☒ | ☐ Animals and other organisms |
| ☒ | ☐ Human research participants |
| ☒ | ☐ Clinical data |
| ☒ | ☐ Dual use research of concern |

## Methods

| n/a | Involved in the study |
|---|---|
| ☒ | ☐ ChIP-seq |
| ☐ | ☒ Flow cytometry |
| ☒ | ☐ MRI-based neuroimaging |

# Antibodies

| | |
|---|---|
| Antibodies used | Primary antibodies: CENP-A (2048S, Cell Signaling, clone C51A7, lot:4), H3K9me3 (07-523, Millipore, lot:2793831) or (07-442, Millipore), H3K27me3 (Ab6002, Abcam, clone mAbcam 6002, lot: 3018864) or (07-449, Millipore), Esrrb (PP-H6705-00, Perseus Proteomics, clone H6705), Sox2 (ab97959, Abcam), H3S10ph (ab5176, Abcam), and Histone H3 (Active Motif 61476, clone 1B1-B2). Secondary antibodies: anti-mouse-Alexa488 (A11001, Invitrogen), anti-rabbit-Alexa488 (A11008, Invitrogen), anti-mouse-Alexa568 (A11031, Invitrogen), anti-rabbit-Alexa680 (A21109, Invitrogen), anti-mouse-Alexa790 (A11371, Invitrogen). |
| Validation | CENP-A (2048S):<br>-Supplier website: (C51A7) Rabbit mAb detects endogenous levels of total mouse CENP-A protein. This antibody does not cross-react with other histone proteins, including Histone H3.<br>-Validated for IF (Smoak et al, Current Biology, 2016).<br><br>H3K9me3 Abs:<br>-07-523, supplier website: Recognizes Histone H3 containing trimethyl-lysine 9 and, to a lesser extent, dimethyl-lysine 9. MW ~17 kDa. A broad species cross-reactivity is expected.<br>-07-442, supplier website: Specificity=Trimethyl-histone H3 (Lys9). Validation includes Immunocytochemistry. Broad species cross-reactivity expected, including mouse.<br>-both validated for IF using Suv39h1/h2 double knockout cells (Djeghloul et al, Stem Cell Reports 2016 and this study).<br><br>H3K27me3 Abs:<br>-Ab6002, supplier website: This antibody is specific for histone H3 tri-methylated at K27. Suitable for ICC/IF. Reacts with mouse.<br>-07-449, supplier website: Specificity=Trimethylated histone H3 (Lys27) (dot blot tested). Reacts with mouse. Validated in ICC.<br>-both validated for IF using EED Knockout ESCs (Djeghloul et al, Nature Communications 2020, this study, unpublished data).<br><br>Esrrb (PP-H6705-00):<br>-Supplier specification sheet: This antibody specifically recognizes human ERR beta (ESRRB) and cross reacts with mouse and rat ERR beta. This antibody does not recognize human ERR alpha and gamma. Tested for western blot, immunoprecipitation and immunohistochemistry.<br>-Validated for ChIP, IF, and WB using Esrrb KO cells (EKOiE ES cells) (Festuccia et al, Nature Cell Biology 2016; Festuccia et al, The Embo Journal 2018, Festuccia et al, Genome Res 2019)<br><br>Sox2 (ab97959):<br>-Supplier website: tested for ICC/IF, reacts with mouse.<br>-Validated for IF (Percharde et al, Genes Dev, 2012; Djeghloul et al, Nature Communications 2020)<br><br>H3S10ph (ab5176):<br>-Supplier website: Specificity - This antibody is specific for phosho S10 of histone H3. We believe that it does not recognise the non-modified histone - no blocking is seen with the non-phospho peptide. Predicted to work with mouse. Tested for ICC.<br>-Validated for IF using interphase and metaphase-arrested cells (this study).<br><br>H3 (61476):<br>-Supplier website: validated for WB, wide range of species reactivity predicted. |

# Eukaryotic cell lines

Policy information about cell lines

| | |
|---|---|
| Cell line source(s) | Mouse ESCs used in this study: WT ESC (WT26), Suv39h dn (DN57), Suv39h1-EGFP (Suv39h dn ESCs overexpressing full length Suv39h1), from Thomas Jenuwein's Lab (Lehnertz et al, Curr Biol 2003; Velazquez Camacho et al, Elife 2017), and Esrrb-tdTomato ESCs (gift from Nicola Festuccia; Fesrtuccia et al, Nature Cell Biology 2016).<br>Mouse Embryonic Fibroblasts (MEFs) used in this study were WT (W8), Suv39h dn (D5), WT (clone Eset25 control), Suv39h1/2 -/- (CRISPR clone B1), Setdb1/2 -/- (CRISPR clone A4+OHT) and G9a/Glp -/- (CRISPR clone H7), from Thomas Jenuwein's Lab (Peters et al, Cell 2001; Montavon et al, Nat Commun 2021). |
| Authentication | All cell lines were tested for Karyotype, genotyped using appropriate primers, and validated by WB and immunofluorescence. |
| Mycoplasma contamination | All cell lines were tested negative for mycoplasma contamination. |

| Commonly misidentified lines | No commonly misidentified cell lines were used in this study. |
|---|---|
| (See ICLAC register) | |

# Flow Cytometry

## Plots

Confirm that:

☒ The axis labels state the marker and fluorochrome used (e.g. CD4-FITC).

☒ The axis scales are clearly visible. Include numbers along axes only for bottom left plot of group (a 'group' is an analysis of identical markers).

☒ All plots are contour plots with outliers or pseudocolor plots.

☒ A numerical value for number of cells or percentage (with statistics) is provided.

## Methodology

| | |
|---|---|
| Sample preparation | Chromosome sorting:<br>Chromosomes were extracted from the different cell lines and stained with Hoechst 33258 and Chromomycin A3. Chromosomes were examined by flow cytometry using a Becton Dickinson Influx equipped with spatially separated lasers. Hoechst 33258 was excited using a (Spectra Physics Vanguard, air cooled) 355 nm laser with a power output of 350 mW. Hoechst 33258 fluorescence was collected using a 400 nm long pass filter in combination with a 500 nm short pass filter. Chromomycin A3 was excited using a (Coherent Genesis, water cooled) 460 nm laser with a power output of 500 mW. Chromomycin A3 fluorescence was collected using a 500 nm long pass filter in combination with a 600 nm short pass filter. Forward scatter was measured using a (Coherent Sapphire) 488 nm laser with a power output of 200 mW and this was used as the trigger signal for data collection. Chromosomes were sorted at an event rate of 20000 per second. A 70-micron nozzle tip was used along with a drop drive frequency set to ~96 KHz and the sheath pressure was set to 65 PSI.<br><br>PI staining:<br>Cells were fixed with ice-cold 70% ethanol, washed twice with PBS and resuspended in staining buffer containing 0.05 mg/ml of PI, 1 mg/ml RNaseA, and 0.05% NP40. Samples were incubated for 10 min at room temperature (RT) and 20 min on ice. |
| Instrument | Chromosomes: Becton Dickinson Influx equipped with spatially separated air cooled lasers.<br>PI staining: BD Fortesa flow cytometer. |
| Software | BD FACS software (v1.2.0.142, Influx); BD DIVA (v8.0.1, Fortesa). |
| Cell population abundance | Gating strategy and percentages of chromosome 19 and X is provided in the manuscript.<br>Purity of individual chromosome sort was assessed by DNA FISH with mouse chromosome 19- or X-specific paints. 99-100% sample purity was achieved.<br>Gating strategy and percentages of each phase of cell cycle are provided in the manuscript. |
| Gating strategy | Chromosomes were first gated on a plot of high Hoechst 33258 vs low Forward scatter signal to gate out debris and clumps. This first gate was then used to create a chromosome karyotype by plotting Hoechst 33258 vs Chromomycin A3 fluorescence. Gating strategies for chromosome sorting and cell cycle analysis are provided in the manuscript. |

☒ Tick this box to confirm that a figure exemplifying the gating strategy is provided in the Supplementary Information.

