## [Peer review file. · Nature Structural & Molecular Biology]

Peer Review Information

Manuscript Title: Loss of H3K9 tri-methylation alters chromosome compaction and transcription factor retention during mitosis

Corresponding author name(s): Amanda (G.) Fisher, Dounia Djegloul

Reviewer Comments & Decisions:

Decision Letter, initial version:
--

Message: 12th May 2022

Dear Professor Fisher,

Thank you again for submitting your manuscript "Loss of H3K9 tri-methylation alters chromosome compaction and transcription factor retention during mitosis". I apologize for the delay in responding, it took longer than expected to receive the full set of referee reports needed to make a full and fair decision on your study. We now have comments (below) from the 3 reviewers who evaluated your paper. In light of those reports, we remain interested in your study and would like to see your response to the comments of the referees, in the form of a revised manuscript.

You will see that although all referees find the study of potential interest, they note the need for additional data and analysis to support the model that H3K9 tri-methylation is important for bona fide 'bookmarking'. Please be sure to address/respond to all concerns of the referees in full in a point-by-point response and highlight all changes in the revised manuscript text file.

We appreciate the requested revisions are extensive. We thus expect to see your revised manuscript within 6 months. If you cannot send it within this time, please let us know. We will be happy to consider your revision as long as nothing similar has been accepted for publication at NSMB or published elsewhere. Should your manuscript be substantially delayed without notifying us in advance and your article is eventually published, the received date would be that of the revised, not the original, version.

Reporting Summary:

Please note that all key data shown in the main figures as cropped gels or blots should be presented in uncropped form, with molecular weight markers. These data can be aggregated into a single supplementary figure. While these data can be displayed in a relatively informal style, they must refer back to the relevant figures. These data should be submitted with the last revision, prior to acceptance, but you may want to start putting it together at this point.

We require deposition of coordinates (and, in the case of crystal structures, structure factors) into the Protein Data Bank with the designation of immediate release upon

publication (HPUB). Electron microscopy-derived density maps and coordinate data must be deposited in EMDb and released upon publication. Deposition and immediate release of NMR chemical shift assignments are highly encouraged. Deposition of deep sequencing and microarray data is mandatory, and the datasets must be released prior to or upon publication. To avoid delays in publication, dataset accession numbers must be supplied with the final accepted manuscript and appropriate release dates must be indicated at the galley proof stage. Please find the complete NRG policies on data availability at <http://www.nature.com/authors/policies/availability.html>.

<https://mts-nsmb.nature.com/cgi-bin/main.plex?el=A2J2Ct1A7Nhi3J3A9ftdbTQbxAUAsj24tfzGJc4o3wZ>

Sincerely,

Carolina

Carolina Perdigoto, PhD
Chief Editor
Nature Structural & Molecular Biology
orcid.org/0000-0002-5783-7106

Referee expertise:

Referee #1: chromatin biology, bookmarking

Referee #2: chromatin biology, epigenetic memory

Referee #3: chromatin biology, proteomics

Reviewers' Comments:

Reviewer #1:

Remarks to the Author:

Review of "Loss of H3K9 tri-methylation alters chromosome compaction and transcription factor retention during mitosis"

Summary:

In this paper, Djeghloul et al. investigate the effect of depleting H3K9me3 by deleting Suv39h1/2 on the protein content of mitotic chromosomes. During mitosis, several chromatin-bound proteins, notably pluripotent transcription factors, remain associated with mitotic chromosomes. This is referred to as mitotic bookmarking, and has typically been assessed by microscopy and ChIP. Here, the authors isolate native mitotic chromosomes by flow-cytometry. They first carry out microscopy analysis of Chr X and 19, and then proteomic analysis (LC MS/MS) of sorted chromosomes to identify proteins enriched on mitotic chromosomes. This experiment is carried out with wild type and Suv39h1/2 knockout cells (as well as cells lacking other H3K9 methyltransferases, which do not have the same effects), and both in mESCs and MEFs (representing differentiated cells). The method was previously used by this team to demonstrate that proteins involved in gene repression such as PRC1, PRC2, Suv39h1 and Suv39h2 are enriched on mitotic chromosomes.

The first key finding is that

depleting Suv39h1 and Suv39h2 and thus removing H3K9me3, increases chromosome and centromere compaction during mitosis and increases H3K27me3 and H3S10ph on mitotic chromosomes. This surprising observation can be explained by the replacement of H3K9me3 by H3K27me3 in absence of Suv39h1/Suv39h2 (consistent with previous work showing antagonism between H3K9me3 and H3K27me3 deposition). This was confirmed by deletion and chemical inhibition of Suv39h1/Suv39h2, and by the resupply of Suv39h1/Suv39h2 which reversed mitotic chromosome compaction and deposition of H3K27me3 at centromeric regions. Thus, PRC2 compensates for the loss of H3K9me3, especially in the centromeric regions, but results in hyper compaction of chromosomes. Notably, chemical inhibition of PRC2 in Suv39h1/Suv39h2 depleted cells, resulted in bigger chromosomes and centromeres. The authors also show that chromatin accessibility is not strongly affected in the absence of H3K9me3 (using ATAC seq), although this is not investigated in detail.

The second main finding, from the proteomic analysis, is that specific factors are depleted from mitotic chromosomes in the mutant versus wild type, including transcription factors such as Esrrb that were previously implicated in mitotic bookmarking. In Suv39h1/Suv39h2 depleted cells, inhibition of PRC2 (H3K27me3) and Aurora Kinase (H3S10ph) rescued the retention of Tead4 and Esrrb respectively, demonstrating that different modifications affect retention of these factors. These data reveal a novel role for H3K9me3/heterochromatin in shaping the protein content of mitotic cells. The obvious limitation of the study is that nothing is determined about the mechanism underlying the change in binding of transcription factors; indeed, it is not even clear if the factors are lost specifically from bookmarked sites, or if it is non-specific binding (which may also be functionally significant) is lost. In my view, the paper makes a large advance, and does so in a careful and rigorous way. I believe the mechanistic follow up would be a full study on its own. There are a few points that should be addressed.

Major points:

1) What is the sex of the cell lines used? Given that the authors analyze the X-

chromosome, this is highly relevant—if the cells are female, how do they deal with possibly heterogeneity due to active and inactive X chromosomes?

2) The authors focus on the change in protein binding in mitosis, and argue that because there does not seem to be a large change in expression levels in the knockout cells, the effects are specific to mitosis. However, they do not provide any evidence that these proteins are normally bound in interphase. At least for *Esrrb*, they should also analyze binding to interphase chromatin. It is possible that there are already defects in binding in interphase, that then carry through to mitosis. This does not invalidate the results in mitosis, but is an important mechanistic distinction if the effect is restricted to mitosis (as implied by the authors), or accompanied (and possibly partially caused by) effects in interphase. Presumably the changes in histone PTMs, which may drive changes in protein binding in mitosis, are also present in interphase. This point needs to be addressed in the text/presentation of the MS results as well.

Minor points:

- In Figure 2G, the color of sorted chromosomes *Suv39h dn* should be brighter.
- Line 174 (and all places where total protein numbers are stated), why do these numbers not match the sum of the numbers on the volcano plots?
- In Fig 4C, the general level of *Esrrb* in *Chaetocin* treated cells seems decreased compared to non-treated cells. To compare between treated and non-treated cells, the ratio of *Esrrb* signal on mitotic chromosome vs cytoplasm might be more accurate. (this is also relevant for point 2 above)
- I may have missed it, but I did not find information about availability of the proteomic and ATAC-seq data

Outstanding features:

This is a technically solid paper. The results are carefully controlled, and presented in a quantitative manner. The results are clear and compelling. Another strength is the use of both mESCs and MEFS, since PRC2 function is distinct in stem cells and differentiated cells. The paper is well written, and both the introduction and discussion are scholarly. The discussion nicely explains what could be the potential role of some proteins enriched on mitotic chromosomes relatively to the mitotic bookmarking model.

New highlights in the field:

The mitotic bookmarking model has been controversial in the field, notably due to the fact that different results are obtained for whether proteins are retained based on technical aspects of experiments (notably fixation). Thus, identification of several chromatin-bound proteins on native isolated mitotic chromosomes by proteomic analyses (along with their previous paper establishing the method) is a novel and original approach to the problem. The authors are also pioneering the contribution of histone post-translational modifications in the retention of proteins on mitotic chromosomes. They notably showed that each histone modification could selectively play a role in retaining pluripotent transcription factors on mitotic chromosomes. Thus, this article is of immediate interest for the mitotic bookmarking field and the general fields of mitosis/cell cycle, transcription, and epigenetics. As noted above, while these new data are interesting, the mechanisms behind the selective retention of proteins on mitotic chromosomes by specific histone modifications or their corresponding enzyme remain to be determined. This work opens the door to more in depth analysis.

Reviewer #2:

Remarks to the Author:

In this manuscript, Djeghloul et al explore the role of the Suv39h1/h2 and H3K9me3 modification on mitotic chromosome architecture and on factors that bind to chromosomes during mitosis. Using knock out cells for the methyltransferases Suv39h1/h2, the authors show that loss of H3K9me3 is associated with smaller, more compact, mitotic chromosomes. Loss of H3K9me3 was also associated with an encroachment H3K27me3 (an alternative repressive modification deposited by Polycomb Repressive Complex 2) into centromeric regions, together with elevated H3S10 phosphorylation. The authors then show that inhibiting PRC2 activity in Suv39 double KO cells is sufficient to restore normal mitotic chromosome size, suggesting that in the absence of Suv39h1/h2, aberrant compaction of mitotic chromosomes is driven, at least in part, by H3K27me3. Interestingly, however, the authors demonstrate that in the absence of Suv39h1/h2, chromatin accessibility throughout the genome (as measured by ATAC-seq) is largely unchanged. Next, the authors use LC-MS to look at proteins bound to mitotic chromosomes in either wildtype or Suv39h1/h2 KO cells. This reveals a group of nearly 300 proteins, associated with functions that include transcription and chromosome organisation, that are lost from mitotic chromosomes in the absence of Suv39h1/h2. Proteins lost from mitotic chromosomes in the absence of Suv39h1/h2 included Esrrb (a well-characterised mitotic bookmarking factor), Tead4 and Tbx3. The authors further characterize the association of these proteins with mitotic chromosomes using various inhibitor and rescue experiments. These further experiments reveal that, while H3K9me3 is important for binding to mitotic chromosomes, the presence of H3K27me3 and H3S10ph exert more context-specific and selective protein-specific effects.

Overall, this is a nice study with very high quality data obtained using cutting-edge experimental approaches. However, I was left with the slight feeling after reading the manuscript carefully that the observations described in the paper did not fully support (or were not fully developed in supporting) the central claim of the study, that H3K9 trimethylation is important for mitotic bookmarking. I have expanded on these points in more detail below:

1) The authors propose a model in which H3K9me3 is required for mitotic bookmarking by key transcription factors. However, I was not entirely compelled by the authors' evidence for mitotic bookmarking, and how it is influenced by H3K9me3. The central premise of bookmarking is that the binding of certain transcription factors is retained at key regulatory elements during mitosis to enable timely gene regulation after exit from mitosis. In the present study, the authors use bulk (LC-MS) or low resolution (immunofluorescence) chromatin association analysis techniques as an indicator of mitotic bookmarking. However, I was unsure if the authors were really examining transcription factors engaged with their specific binding sites on mitotic chromosomes (i.e. true bookmarking)? Or is this a separate TF pool associated with distinct chromosome regions? Without better knowledge of where these TFs are bound during mitosis and how this binding changes in the absence of Suv39h1/h2 and H3K9me3, it is difficult to ascertain whether this is really 'bookmarking' as the authors have invoked.

2) If Suv39h1/h2 and H3K9me3 are indeed required for mitotic bookmarking by key transcription factors, what effect does this have on transcription in dividing cells? Presumably loss of Suv39h1/h2 and H3K9me3 would cause a delay in the re-

establishment of transcription following exit from mitosis?

3) The increased compaction of mitotic chromosomes in the absence of H3K9me3 is an interesting observation. Is this linked to transcription factor retention during mitosis? The fact that compaction is ablated by PRC2 inhibition, suggests PRC2/K27me3 (which spread aberrantly when K9me3 was removed) may be responsible for this compaction (although the authors later suggest Histone H1 may contribute based on their proteomic data). Where does PRC2/K27me3 spread to? Is this spatially related to TF binding in these regions?

Reviewer #3:

Remarks to the Author:

How nuclear architecture and especially heterochromatic domains are maintained during mitosis is largely unclear. Djeghloul et al set out to study the effect of heterochromatic H3K9me3 on mitotic chromosomes and chromosome-protein interactions during mitosis. They do this by isolating mitotic chromosomes of WT or Suv39h1 double null ESCs or MEFs and studying chromosome and centromeric architecture as well as performing interaction proteomics on mitotic chromosomes. Mitotic chromosomes of cells lacking Suv39h1 (and therefore H3K9me3) appeared smaller and showed increased compaction at centromeres, showed decreased centromeric H3K9me3 intensity, which was compensated for by increased PRC2-dependent H3K27me3. Importantly, chromatin accessibility was not affected in Suv39 dn cells. Through proteome analysis of mitotic lysates and sorted mitotic chromosomes, factors were identified that were either enriched or depleted on mitotic chromosomes. Specifically, some chromosome binding factors/pluripotency-associated proteins such as Esrrb were lost in Suv39h dn mitotic chromosomes, and for Esrrb the authors show that this can be rescued by restoring Suv39h1 levels. Unfortunately, however, the molecular mechanism underlying this observation, remain unclear. Surprisingly, decreased retention of Esrrb did not result in loss of chromatin accessibility at specific Esrrb sites. Altogether, the authors present interesting observations that are worth reporting, although these observations are largely descriptive. Furthermore, there are a number of issues that need to be addressed prior to publication:

Major comments:

1. In suppl. Figure 1b, Suv39h1 and Suv39h1 are found enriched on mitotic chromosomes, can this also be shown by IF, does Suv39h1 specifically localize towards centromeric regions, similar to H3K9me3 (in figure 2a)?
2. In supplemental figure S1d, it seems that chromosome size (not centromere size) is even slightly increased in the absence of Setdb1/2 and G9a/Glp. Is this a statistically significant effect, and do the authors have any explanation for this? What are the levels of H3K9me3 (and me1/2) in the Setdb1/2 and G9a/Glp knock-out clones used? If H3K9me1/2 levels are decreased compared to control, would this not result in lower levels of H3K9me3 as well, due to less substrate?
3. Following up on this, in figure S2c it seems that H3K9me3 levels are even increased in Setdb1/2 and G9a/Glp knockout clones. Can significance be added to this figure, and is this expected? Could this therefore have the opposite effect of Suv39h loss on chromosome size?
4. What are the consequences of Suv39h depletion on mitotic progression?
5. The volcano plots (i.e Figure 3) show significant proteins with a moderate ratio (less than 2-fold). The authors should indicate why these cut-offs were chosen. In certain plots (i.e Figure 4A) there are more outliers than background proteins (although numbers as in

- Figure 3 are not shown in Figure 4). The statistical filtering of the proteomics data should be further motivated and described. In my opinion, stricter cut-offs should be chosen given the broad distribution of the data points
6. The GO terms in Figure 3 are not very useful in my opinion
 7. The experiments in figure 5a nicely show a rescue of the phenotype. How do the levels of Suv39h shown compare to WT conditions?
 8. Does depletion of Esrrb now affect chromosome size or centromere size?
 9. In the discussion, the authors describe that Suv39h1/2 deficient mice show aneuploidy with increased tumour incidence, and that cells losing H3K9me3 experience genomic instability. What are the cellular consequences of H3K9me3 loss and the subsequent Esrrb decrease at mitotic chromosomes? Does Suv39h depletion affect mitotic progression or correct chromosome segregation during mitosis? Does increased chromosome and centromere size observed upon Suv39h loss affect mitosis? And does Esrrb loss by itself affect mitosis and/or cell division?
 10. If the H3K9me3-levels are important for genomic bookmarking, and Esrrb is one of the factors lost, does bookmarking indeed not function properly in the next G1-phase? So are Esrrb-dependent transcripts lost in H3K9me3-deficient cells? Do you actually find a 'bookmarking-deficiency'?
 11. Nowadays instead of working with KO versus WT cells one would probably use a complementary degron approach to perturb Suv39h and then study the functional consequences. Do the authors observe similar findings when transiently perturbing/depleting Suv39h?

Minor comments:

1. For clarity, I would consider rephrasing some of the sentences such as sentence 101-102 ('chromosomes released'), 118-120 ('specifically pairwise deletions of..'),
2. What other chromatin modifications are the authors referring to in line 145 ('compensatory increase in H3K27me3 (or other chromatin modifications) rather than loss of H3K9me3 per se..')? Is this referring to DNA methylation?
3. In figure 3a and d, it seems as if there are 4468 (WT MEFs) and 4519 (Suv39h dn MEFs) proteins detected, rather than the 5789 and 5784 proteins mentioned in the text. Can the authors comment on these differences in number? Also, the numbers displayed in the volcano plots (3a, 3d) on the right (the proteins enriched on sorted chromosomes) do not completely match with the numbers shown in the Venn-diagrams (3b and 3e)
4. It appears as if the black line showing the threshold p-value in figure 4a is not at the same height/cutoff when comparing the left and right vulcanoplots.
5. Can the authors discuss in more detail the factors that are lost from mitotic chromosomes in Suv39h ko cells in their MS experiments?
6. Are there any non-histone targets of Suv39h that could impact mitotic chromosomes?

Author Rebuttal to Initial comments

Reviewer #1:

Remarks to the Author:

Review of "Loss of H3K9 tri-methylation alters chromosome compaction and transcription factor retention during mitosis"

Summary:

In this paper, Djeghloul et al. investigate the effect of depleting H3K9me3 by deleting Suv39h1/2 on the protein content of mitotic chromosomes. During mitosis, several chromatin-bound proteins, notably pluripotent transcription factors, remain associated with mitotic chromosomes. This is referred to as mitotic bookmarking, and has typically been assessed by microscopy and ChIP. Here, the authors isolate native mitotic chromosomes by flow-cytometry. They first carry out microscopy analysis of Chr X and 19, and then proteomic analysis (LC MS/MS) of sorted chromosomes to identify proteins enriched on mitotic chromosomes. This experiment is carried out with wild type and Suv39h1/2 knockout cells (as well as cells lacking other H3K9 methyltransferases, which do not have the same effects), and both in mESCs and MEFs (representing differentiated cells). The method was previously used by this team to demonstrate that proteins involved in gene repression such as PRC1, PRC2, Suv39h1 and Suv39h2 are enriched on mitotic chromosomes.

The first key finding is that depleting Suv39h1 and Suv39h2 and thus removing H3K9me3, increases chromosome and centromere compaction during mitosis and increases H3K27me3 and H3S10ph on mitotic chromosomes. This surprising observation can be explained by the replacement of H3K9me3 by H3K27me3 in absence of Suv39h1/Suv39h2 (consistent with previous work showing antagonism between H3K9me3 and H3K27me3 deposition). This was confirmed by deletion and chemical inhibition of Suv39h1/Suv39h2, and by the resupply of Suv39h1/Suv39h2 which reversed mitotic chromosome compaction and deposition of H3K27me3 at centromeric regions. Thus, PRC2 compensates for the loss of H3K9me3, especially in the centromeric regions, but results in hyper compaction of chromosomes. Notably, chemical inhibition of PRC2 in Suv39h1/Suv39h2 depleted cells, resulted in bigger chromosomes and centromeres. The authors also show that chromatin accessibility is not strongly affected in the absence of H3K9me3 (using ATAC seq), although this is not investigated in detail.

The second main finding, from the proteomic analysis, is that specific factors are depleted from mitotic chromosomes in the mutant versus wild type, including transcription factors such as Esrrb that were previously implicated in mitotic bookmarking. In Suv39h1/Suv39h2 depleted cells, inhibition of PRC2 (H3K27me3) and Aurora Kinase (H3S10ph) rescued the retention of Tead4 and Esrrb respectively, demonstrating that different modifications affect retention of these factors. These data reveal a novel role for H3K9me3/heterochromatin in shaping the protein content of mitotic cells. The obvious limitation of the study is that nothing is determined about the mechanism underlying the change in binding of transcription factors; indeed, it is not even clear if the factors are lost specifically from bookmarked sites, or if it is non-specific binding (which may also be functionally significant) is lost. In my view, the paper makes a large advance, and does so in a careful and rigorous way. I believe the mechanistic follow up would be a full study on its own. There are a few points that should be addressed.

We are grateful to the reviewer for their encouraging comments.

Major points:

1) What is the sex of the cell lines used? Given that the authors analyze the X-chromosome, this is highly relevant—if the cells are female, how do they deal with possibly heterogeneity due to active and inactive X chromosomes?

Male cells (MEFs and ESCs) were used throughout this study so as to minimize heterogeneity; male parental and H3K9 methylation-mutant cells were obtained from the Jenuwein Laboratory and have been extensively characterised previously (Lehnertz *et al.*, 2003; Peters *et al.*, 2001; Mantavon *et al.*, 2021). In our study FISH analyses confirmed that individual *Suv39h dn* and WT ESCs cells contained two chromosomes 19 and a single X chromosome (Figure 1g and 1h), consistent with being male. The *Esrrb-tdTomato* ESC line was a kind gift from Nicola Festuccia and is derived from a male parental E14Tg2a ES line (Festuccia *et al.*, 2016). We have included this information in the revised materials and methods section.

2) The authors focus on the change in protein binding in mitosis, and argue that because there does not seem to be a large change in expression levels in the knockout cells, the effects are specific to mitosis. However, they do not provide any evidence that these proteins are normally bound in interphase. At least for *Esrrb*, they should also analyze binding to interphase chromatin. It is possible that there are already defects in binding in interphase, that then carry through to mitosis. This does not invalidate the results in mitosis, but is an important mechanistic distinction if the effect is restricted to mitosis (as implied by the authors), or accompanied (and possibly partially caused by) effects in interphase. Presumably the changes in histone PTMs, which may drive changes in protein binding in mitosis, are also present in interphase. This point needs to be addressed in the text/presentation of the MS results as well.

This is a very good point and additional experiments were performed to investigate this; we examined *Esrrb* binding by ChIP in asynchronous/interphase and mitotic WT and *Suv39h dn* ESCs. We selected candidate sites from prior publications (Festuccia *et al.*, 2016) as representing sites that were either bookmarked throughout mitosis by *Esrrb* (*Capn2*, *Esrrb*, *Jam2-s1*, *Jam2-s2*, *Tbx3*, *Tet2*), bound by *Esrrb* only in interphase (*Mgat3*, *Twistnb*) or control sites that do not bind *Esrrb* (*Esrrb-3'*, *Actb*). As shown in new Figure 4c, we observed a statistically significant decrease in *Esrrb* binding at five of seven bookmarked sites in *Suv39h dn* ECS versus WT, exclusively in mitotic samples (new Figure 4c, upper panel). In asynchronous/interphase samples (new Figure 4c, lower panel), *Esrrb* binding appeared slightly lower in *Suv39h dn* mutants, but these decreases were not statistically significant. In the absence of genome-wide data, we cannot however completely rule out that loss of H3K9me3-loss causes defects in interphase binding that could carry through to mitosis.

We certainly agree that differences in histone PTMs may be present in interphase (discussed in the manuscript). This is for example the case in H3K27me3 recruitment to pericentric domains in the absence of H3K9me3, which has already been reported for *Suv39h dn* mutant cells in interphase (Peters *et al.*, 2003; Puschendorf *et al.*, 2008; Saksouk *et al.*, 2014). Other PTMs, such as increased H3S10ph seen on *Suv39h dn* chromosomes are much more likely to be specific to mitosis - particularly as H3S10ph and the adjacent H3K9me3 can compete (Peng *et al.*, 2018); and H3S10ph is increased during mitosis. In this setting, increased H3S10ph and an absence of H3K9me3, could drive changes in (Esrrb) protein binding in mitosis; the observation that treatment of *Suv39h dn* ESCs with Hesperadin, an aurora B kinase inhibitor that reduces H3S10ph levels in mitosis, effectively restores Esrrb enrichment at mitotic chromosomes (Figure 5d) is consistent with this interpretation.

Minor points:

- In Figure 2G, the color of sorted chromosomes *Suv39h dn* should be brighter.

We changed the colour of the *Suv39h dn* sorted chromosomes (ATAC-Seq profile).

- Line 174 (and all places where total protein numbers are stated), why do these numbers not match the sum of the numbers on the volcano plots?

We apologise for any confusion caused. The total protein numbers stated in the manuscript were the number of protein hits detected in the mitotic lysate samples after filtering steps of raw MaxQuant data in Perseus. Volcano plots display proteins that are present in both mitotic lysate and sorted chromosome samples, excluding the protein hits that are present in only one condition (using the robust filtering and statistical analysis method that we have described previously in Djeghloul *et al.*, 2020). We have clarified this in the revised narrative and added details on the approach and filtering steps to the materials and methods section. We have also included in supplemental Tables 1 and 2 (tab2) the total number of proteins identified in each condition (mitotic lysates and sorted chromosomes) and number of proteins shared between the lysate and sorted chromosome samples (this number corresponds to the total number of hits displayed in the volcano plots).

- In Fig 4C, the general level of Esrrb in Chaetocin treated cells seems decreased compared to non-treated cells. To compare between treated and non-treated cells, the ratio of Esrrb signal on mitotic chromosome vs cytoplasm might be more accurate. (this is also relevant for point 2 above)

We re-examined Esrrb levels in Chaetocin-treated versus non-treated (DMSO), as requested, by quantifying Esrrb signal intensity in treated and non-treated cells in interphase and in mitosis,

using live cell imaging. In interphase cells, we saw no statistically significant differences in Esrrb-tdTomato signal levels following treatment with chaetocin versus DMSO. This result is incorporated into a revised Figure (new Figure 4d, right-hand plot). In mitotic cells we saw a significant difference in Esrrb-tdTomato mean signal intensity in Chaetocin treated versus control samples (new Figure 4d, left-hand plot). We also calculated the ratio of Esrrb-tdTomato signal intensity on mitotic chromosomes versus cytoplasm, as the reviewer suggested using the live imaging data in Figure 4d. This calculation produces a similar result showing a significant reduction in mitotic Esrrb signal following Chaetocin treatment.

Figure: Ratio of Esrrb-tdTomato mean intensity signal on mitotic chromosomes vs cytoplasm for each condition using live-cell imaging of Esrrb-tdTomato ESCs pre-treated with DMSO or 100 nM of Chaetocin. Mean \pm SD is shown, $n = 30$ mitotic cells analysed.

To exclude that Chaetocin treatment (and Suv39h1 inhibition) results in a generalised decrease in transcription factor retention on mitotic chromosomes (ie. is non-specific), we sorted chromosomes 19 and X from Esrrb-tdTomato ESCs that had been pre-treated with either Chaetocin or DMSO, and then examined Esrrb and Sox2 levels. A significant reduction in Esrrb signal was observed in Chaetocin-treated chromosomes (new Figure S4g) as anticipated, while levels of Sox2 remained unaffected by Chaetocin treatment (new Figure S4h).

- I may have missed it, but I did not find information about availability of the proteomic and ATAC-seq data

These were provided in the data availability section line 657 (page 25 of the revised manuscript).

Outstanding features:

This is a technically solid paper. The results are carefully controlled, and presented in a quantitative manner. The results are clear and compelling. Another strength is the use of both mESCs and MEFS, since PRC2 function is distinct in stem cells and differentiated cells. The paper is well written, and both the introduction and discussion are scholarly. The discussion nicely explains what could be the potential role of some proteins enriched on mitotic chromosomes relatively to the mitotic bookmarking model.

New highlights in the field:

The mitotic bookmarking model has been controversial in the field, notably due to the fact that different results are obtained for whether proteins are retained based on technical aspects of experiments (notably fixation). Thus, identification of several chromatin-bound proteins on native isolated mitotic chromosomes by proteomic analyses (along with their previous paper establishing the method) is a novel and original approach to the problem.

The authors are also pioneering the contribution of histone post-translational modifications in the retention of proteins on mitotic chromosomes. They notably showed that each histone modification could selectively play a role in retaining pluripotent transcription factors on mitotic chromosomes. Thus, this article is of immediate interest for the mitotic bookmarking field and the general fields of mitosis/cell cycle, transcription, and epigenetics. As noted above, while these new data are interesting, the mechanisms behind the selective retention of proteins on mitotic chromosomes by specific histone modifications or their corresponding enzyme remain to be determined. This work opens the door to more in depth analysis.

We are very pleased that the referee considers our work novel and of immediate interest.

Reviewer #2:

Remarks to the Author:

In this manuscript, Djeghloul et al explore the role of the Suv39h1/h2 and H3K9me3 modification on mitotic chromosome architecture and on factors that bind to chromosomes during mitosis. Using knock out cells for the methyltransferases Suv39h1/h2, the authors show that loss of H3K9me3 is associated with smaller, more compact, mitotic chromosomes. Loss of H3K9me3 was also associated with an encroachment H3K27me3 (an alternative repressive modification deposited by Polycomb Repressive Complex 2) into centromeric regions, together with elevated H3S10 phosphorylation. The authors then show that inhibiting PRC2 activity in Suv39 double KO cells is sufficient to restore normal mitotic chromosome size, suggesting that in the absence of Suv39h1/h2, aberrant compaction of mitotic chromosomes is driven, at least in part, by H3K27me3. Interestingly, however, the authors demonstrate that in the absence of Suv39h1/h2, chromatin accessibility throughout the genome (as measured by ATAC-seq) is largely unchanged. Next, the authors use LC-MS to look at proteins bound to mitotic chromosomes in either wildtype or Suv39h1/h2 KO cells. This reveals a group of nearly 300 proteins, associated with functions that include transcription and chromosome organisation, that are lost from mitotic chromosomes in the absence of Suv39h1/h2. Proteins lost from mitotic chromosomes in the absence of Suv39h1/h2 included Esrrb (a well-characterised mitotic bookmarking factor), Tead4 and Tbx3. The authors further characterize the association of these proteins with mitotic chromosomes using various inhibitor and rescue experiments. These further experiments

reveal that, while H3K9me3 is important for binding to mitotic chromosomes, the presence of H3K27me3 and H3K9me3 exert more context-specific and selective protein-specific effects.

Overall, this is a nice study with very high quality data obtained using cutting-edge experimental approaches. However, I was left with the slight feeling after reading the manuscript carefully that the observations described in the paper did not fully support (or were not fully developed in supporting) the central claim of the study, that H3K9 trimethylation is important for mitotic bookmarking. I have expanded on these points in more detail below:

1) The authors propose a model in which H3K9me3 is required for mitotic bookmarking by key transcription factors. However, I was not entirely compelled by the authors' evidence for mitotic bookmarking, and how it is influenced by H3K9me3. The central premise of bookmarking is that the binding of certain transcription factors is retained at key regulatory elements during mitosis to enable timely gene regulation after exit from mitosis. In the present study, the authors use bulk (LC-MS) or low resolution (immunofluorescence) chromatin association analysis techniques as an indicator of mitotic bookmarking. However, I was unsure if the authors were really examining transcription factors engaged with their specific binding sites on mitotic chromosomes (i.e. true bookmarking)? Or is this a separate TF pool associated with distinct chromosome regions? Without better knowledge of where these TFs are bound during mitosis and how this binding changes in the absence of Suv39h1/h2 and H3K9me3, it is difficult to ascertain whether this is really 'bookmarking' as the authors have invoked.

We propose that H3K9me3 is required for the efficient retention of Esrrb on mitotic chromosomes. To address "bookmarking" specifically, we have examined Esrrb binding in asynchronous (interphase) and mitotic cells by ChIP, using chromatin isolated from WT and *Suv39h dn* ESCs (methods outlined by Festuccia and colleagues in Festuccia *et al.*, 2016 and 2019). We selected candidate sites from prior publications (Festuccia *et al.*, 2016) as representing sites that were either bookmarked in mitosis by Esrrb (*Capn2*, *Esrrb*, *Jam2-s1*, *Jam2-s2*, *Tbx3*, *Tet2*), bound by Esrrb only in interphase (*Mgat3*, *Twistnb*) or control sites that do not bind Esrrb (*Esrrb-3*, *Actb*). As shown in new Figure 4c, we observed a statistically significant decrease in Esrrb binding at five of seven bookmarked sites in *Suv39h dn* ECS versus WT, exclusively in mitotic samples (new Figure 4c, upper panel). In asynchronous/interphase samples (new Figure 4c, lower panel), Esrrb binding appeared slightly lower in *Suv39h dn* mutants, but was not statistically significant.

While this ChIP analysis of a panel of established Esrrb 'bookmarked' targets supports our assertion that H3K9me3-loss negatively impacts Esrrb retention on mitotic chromosomes, we

agree with the referee that we should remain cautious in extrapolating our results – particularly in the absence of genome-wide data. The extreme condensation of chromatin that occurs at mitosis provides opportunities for different physical phenomena such as formation of condensates – aside from factors being engaged at cognate DNA-binding sites – (Price *et al.*, 2022 bioRxiv, <https://www.biorxiv.org/content/10.1101/2022.10.05.511012v1.full.pdf>) and I think that we need to remain open-minded to such possibilities. We have revised the manuscript to better discuss and reflect this viewpoint.

2) If Suv39h1/h2 and H3K9me3 are indeed required for mitotic bookmarking by key transcription factors, what effect does this have on transcription in dividing cells? Presumably loss of Suv39h1/h2 and H3K9me3 would cause a delay in the re-establishment of transcription following exit from mitosis?

Transcript expression in *Suv39h dn* versus WT cells has been reported previously (Bulut-Karslioglu *et al.*, 2014). We now show that the cell cycle profiles and mitotic progression/duration are similar in *Suv39h dn* and WT ESCs (new Figure S2f and S2g) and as requested, we performed mitotic release experiments to investigate whether loss of *Suv39h1/h2* and H3K9me3 compromised primary transcript expression in cells exiting mitosis. As shown below, preliminary analysis of a panel of *bona fide* Esrrb-bookmarked genes suggest that re-expression of some “early-responsive genes” (such as *Klf4* and *Tcfcp2l2*) was delayed in *Suv39h dn* cells as compared to WT, while the expression of others genes regulated by Esrrb “non-early-responsive” (such as *Rex1*, *Nanog*, *Tet2*) was apparently unaffected. More extensive genome-wide data will probably be required to fully understand the transcriptional effects in G1.

Figure: Assessing gene transcription reactivation following mitotic exit into G1. (a) Schematic of experimental strategy used to measure primary transcript expression following release into G1 in WT and *Suv39h dn* ESCs. (b) Representative cell cycle profiles of WT and *Suv39h dn* ESCs following release from CDK1 inhibition was determined by propidium iodide (PI) profile. ESCs were treated with CDK1 inhibitor (RO-3306) for 12h followed by a release. 2 hrs and 3hrs post-release, 70 to 80% of cells re-entered G1 phase. (c) qRT-PCR primary transcript expression analysis of early responsive genes *Tcfcp2l1*, *Klf4*, *Jam2* (Festuccia *et al.*, 2016), other pluripotency genes *Nanog*, *Rex1* or *Esrrb*-regulated gene *Tet2* (Bell *et al.*, 2020) in WT and *Suv39h dn* asynchronous cells and following entry into G1. Mature TBP mRNA was used as a control. Mean + SD is shown, n = 3 independent experiments, P-values of statistically significant changes, measured by unpaired two tailed Student’s t-tests, are indicated.

3)The increased compaction of mitotic chromosomes in the absence of H3K9me3 is an interesting observation. Is this linked to transcription factor retention during mitosis? The fact that compaction is ablated by PRC2 inhibition, suggests PRC2/K27me3 (which spread aberrantly when K9me3 was removed) may be responsible for this compaction (although the authors later suggest Histone H1 may contribute based on their proteomic data). Where does PRC2/K27me3 spread to? Is this spatially related to TF binding in these regions?

Proteomic data show increased levels of histone H1 and H1 variants on sorted mitotic chromosomes from *Suv39h dn* ESCs, as shown in Figure S5, as well as higher levels of H3S10ph on mitotic chromosomes and gain (delocalisation) of H3K27me3 at centromeric regions. Each of

these three features correlates with increased compaction of mitotic chromosomes seen in *Suv39h dn* mitotic chromosomes. We have examined whether reduced retention of *Esrrb* could itself provoke over compaction of mitotic chromosomes and examined two independent models in which *Esrrb* expression can be withdrawn – using Cre-driven Flox recombination (Martello *et al.*, 2012), or using *Esrrb*^{-/-} ESCs engineered to contain a doxycycline inducible *Esrrb* transgene (Festuccia *et al.*, 2016). Removal of *Esrrb* in either model did not alter the size of isolated mitotic chromosomes 19 or X.

Figure: Chromosome 19 and X size analysis following *Esrrb* withdrawal/deletion using two different ESC models.

Regarding where PRC2/H3K27me3 spreads to, we see significant increases in H3K27me3 at the centromere of *Suv39h dn* mitotic chromosomes. We cannot examine H3K27me3 spread genome wide (ie. ChIP-seq) because of the very limited amount of chromatin available from FACS-purified chromosome preps. However, the demonstration that by inhibiting PRC2 activity we can restore TEAD4 retention on H3K9me3-null (*Suv39h dn*) mitotic chromosomes, but not *Esrrb*, and yet inhibition of aurora kinase B rescues *Esrrb* retention but does TEAD4, shows that transcription factor retention through mitosis is dependent upon specific and distinct chromatin features.

Reviewer #3:

Remarks to the Author:

How nuclear architecture and especially heterochromatic domains are maintained during

mitosis is largely unclear. Djeghloul et al set out to study the effect of heterochromatic H3K9me3 on mitotic chromosomes and chromosome-protein interactions during mitosis. They do this by isolating mitotic chromosomes of WT or Suv39h1 double null ESCs or MEFs and studying chromosome and centromeric architecture as well as performing interaction proteomics on mitotic chromosomes. Mitotic chromosomes of cells lacking Suv39h1 (and therefore H3K9me3) appeared smaller and showed increased compaction at centromeres, showed decreased centromeric H3K9me3 intensity, which was compensated for by increased PRC2-dependent H3K27me3. Importantly, chromatin accessibility was not affected in Suv39 dn cells. Through proteome analysis of mitotic lysates and sorted mitotic chromosomes, factors were identified that were either enriched or depleted on mitotic chromosomes. Specifically, some chromosome binding factors/pluripotency-associated proteins such as Esrrb were lost in Suv39h dn mitotic chromosomes, and for Esrrb the authors show that this can be rescued by restoring Suv39h1 levels. Unfortunately, however, the molecular mechanism underlying this observation, remain unclear. Surprisingly, decreased retention of Esrrb did not result in loss of chromatin accessibility at specific Esrrb sites. Altogether, the authors present interesting observations that are worth reporting, although these observations are largely descriptive. Furthermore, there are a number of issues that need to be addressed prior to publication:

Major comments:

1. In suppl. Figure 1b, Suv39h1 and Suv39h2 are found enriched on mitotic chromosomes, can this also be shown by IF, does Suv39h1 specifically localize towards centromeric regions, similar to H3K9me3 (in figure 2a)?

Unfortunately, there are currently no antibodies to Suv39h1 or Suv39h2 that can be used reliably in immunofluorescence assays. Several antibodies perform well in Western blots (such as those developed by the Jenuwein laboratory) but these do not work in immunofluorescence. Despite this, we were able to localise Suv39h1 and Suv39h2 using EGFP-fusion proteins. As shown below, Suv39h1 and Suv39h2 proteins are enriched at centromeres in isolated mitotic chromosomes, similar to the distribution seen for H3K9me3 (in Figure 2a).

Figure: Suv39h1 and Suv39h2 localisation to centromeres in sorted mitotic chromosomes 19 and X.

2. In supplemental figure S1d, it seems that chromosome size (not centromere size) is even slightly increased in the absence of Setdb1/2 and G9a/Glp. Is this a statistically significant effect, and do the authors have any explanation for this?

This is correct and we have added statistics to indicate the size increase. It is possible that in the absence of Setb1/2 and G9a/Glp (and reduced H3K9me1 and H3K9me2) cells are inadvertently selected with compensatory activities (or features) that result in reduced compaction and increased mitotic chromosome size.

What are the levels of H3K9me3 (and me1/2) in the Setdb1/2 and G9a/Glp knock-out clones used? If H3K9me1/2 levels are decreased compared to control, would this not result in lower levels of H3K9me3 as well, due to less substrate?

The levels of H3K9me1, me2 and me3 in the *Setdb1/2*, *G91/Glp* and *Suv39h1/h2* knock-out MEF clones were evaluated previously by Jenuwein and Montavon (Montavon *et al.*, 2021) and are shown below for completeness. Both *Setdb1/2* and *G9a/Glp* knock-out clones do not show a reduction in H3K9me3 (if anything a slight increase).

Figure: Western blot analysis of H3K9 methyltransferase knockout MEFs (from Montavon *et al.*, 2021)

3. Following up on this, in figure S2c it seems that H3K9me3 levels are even increased in *Setdb1/2* and *G9a/Glp* knockout clones. Can significance be added to this figure, and is this expected? Could this therefore have the opposite effect of *Suv39h* loss on chromosome size?

Figure S2c examines H3K9me3 at centromeric domains in mitotic chromosomes isolated from *Setdb1/2*, *G9a/Glp*, or *Suv39h1/h2* knockout MEF clones. H3K9me3 is increased at centromeric (DAPI-bright) regions in *Setdb1/2* or *G9a/Glp* knock-out relative to WT controls and we have added statistical analysis to Fig2c, as requested.

As the reviewer rightly points out, H3K9me3 levels do therefore appear to inversely correlate with mitotic chromosome size. However, we need to be very careful about necessarily inferring any causal relationship here, particularly because we and others (Peters *et al.*, 2003; Puschendorf *et al.*, 2008; Saksouk *et al.*, 2014) have shown that H3K9me3-deficient (*Suv39h dn*) cells acquire H3K27me3, and inhibition of PRC2 can increase mitotic chromosome size. In view of this, commenting on the underlying cause or consequences of *Setdb1/2* or *G9a/Glp* withdrawal on mitotic MEF chromosomes, will require a separate dedicated study.

4. What are the consequences of *Suv39h* depletion on mitotic progression?

This is a very good point. We performed live cell imaging experiments to compare mitotic progression in *Suv39h dn* and WT ESCs and as shown in new Figure S2g, the mitotic profiles and duration are remarkably similar.

We thank the reviewer for this suggestion.

5. The volcano plots (i.e Figure 3) show significant proteins with a moderate ratio (less than 2-fold). The authors should indicate why these cut-offs were chosen. In certain plots (i.e Figure 4A) there are more outliers than background proteins (although numbers as in Figure

3 are not shown in Figure 4). The statistical filtering of the proteomics data should be further motivated and described. In my opinion, stricter cut-offs should be chosen given the broad distribution of the data points

Volcano plots were generated in Perseus using the following 2-sample test parameters to define thresholds: test = t-test; side = both; permutation-based FDR = 0.05 (default); number of randomisations = 250 (default); preserve grouping in randomisations = <none> (default); S0 = 0.1. We have now included details of these parameters in the methods.

Permutation-based FDR provides a robust and reliable estimation of the percentage of proteins mistakenly identified as changing, whilst setting S0 (artificial within groups variance) to a non-zero value allows the fold-change to influence the cut-off in addition to the p-value (Tyanova *et al.*, 2016; Virginia Goss Tusher *et al.*, <https://doi.org/10.1073/pnas.091062498>).

6. The GO terms in Figure 3 are not very useful in my opinion

Opinions are often divided about the usefulness of GO terms, however Figures 3c and 3f illustrate the similarities and overlap in pathways impacted by H3K9 me3-depletion between two very different cell types (pluripotent ESCs versus somatic fibroblasts). This is perhaps unanticipated, and in our view relevant.

7. The experiments in figure 5a nicely show a rescue of the phenotype. How do the levels of Suv39h shown compare to WT conditions?

A full comparison of Suv39h and H3K9me3 levels in WT versus *Suv39h* dn-rescued by either *Suv39h1* or *Suv39h2* has been published previously (Velazquez Camacho *et al.*, 2017) and indicated that Suv39h levels after rescue were not overly increased relative to WT level.

8. Does depletion of Esrrb now affect chromosome size or centromere size?

Esrrb depletion does not affect mitotic chromosome size. We have shown this using two independent models in which Esrrb expression can be withdrawn – using Cre-driven Flox recombination (Martello *et al.*, 2012), or using Esrrb^{-/-} ESCs engineered to contain a doxycycline inducible Esrrb transgene (Festuccia *et al.*, 2016). Removal of Esrrb in either model did not significantly change the size of isolated mitotic chromosomes 19 or X.

Figure: Chromosome 19 and X size analysis following *Esrrb* withdrawal/deletion using two different ESC models.

9. In the discussion, the authors describe that *Suv39h1/2* deficient mice show aneuploidy with increased tumour incidence, and that cells losing H3K9me3 experience genomic instability. What are the cellular consequences of H3K9me3 loss and the subsequent *Esrrb* decrease at mitotic chromosomes? Does *Suv39h* depletion affect mitotic progression or correct chromosome segregation during mitosis? Does increased chromosome and centromere size observed upon *Suv39h* loss affect mitosis? And does *Esrrb* loss by itself affect mitosis and/or cell division?

There is a burgeoning literature on the consequences of H3K9me3 loss in mice, for molecular and cellular processes, and on mis-segregation and genomic instability (Peters *et al.*, 2001; Puschendorf *et al.*, 2008; reviewed in Fadloun *et al.*, 2013; Nicetto & Zaret, 2019). However, in cultured ESCs that are the predominant tool used in our study, we saw no evidence that mitotic progression (duration, karyotype, or profile) was significantly altered in *Suv39h dn*, or following *Esrrb*-depletion.

10. If the H3K9me3-levels are important for genomic bookmarking, and *Esrrb* is one of the factors lost, does bookmarking indeed not function properly in the next G1-phase? So are *Esrrb*-dependent transcripts lost in H3K9me3-deficient cells? Do you actually find a 'bookmarking-deficiency'?

We have examined the impact of H3K9me3-deficiency on *Esrrb*-dependent transcripts in the next G1 phase by performing mitotic release experiments. As shown below, analysis of a panel of *bona fide* *Esrrb*-bookmarked genes showed that the re-expression of some “early-responsive genes” (such as *Klf4* and *Tcfcp212*) was delayed in *Suv39h dn* cells as compared to WT, while the expression of others (such as *Rex1*, *Nanog*, *Tet2*) was unaffected. More extensive genome-wide data will probably be required to fully understand the transcriptional effects in G1.

Figure: Assessing gene transcription reactivation following mitotic exit into G1. (a) Schematic of experimental strategy used to measure primary transcript expression following release into G1 in WT and *Suv39h dn* ESCs. (b) Representative cell cycle profiles of WT and *Suv39h dn* ESCs following release from CDK1 inhibition was determined by propidium iodide (PI) profile. ESCs were treated with CDK1 inhibitor (RO-3306) for 12h followed by a release. 2 hrs and 3hrs post-release, 70 to 80% of cells re-entered G1 phase. (c) qRT-PCR primary transcript expression analysis of early responsive genes *Tcfcp212*, *Klf4*, *Jam2* (Festuccia et al., 2016), other pluripotency genes *Nanog*, *Rex1* or *Esrrb*-regulated gene *Tet2* (Bell et al., 2020) in WT and *Suv39h dn* asynchronous cells and following entry into G1. Mature TBP mRNA was used as a control. Mean + SD is shown, n = 3 independent experiments, P-values of statistically significant changes, measured by unpaired two tailed Student's t-tests, are indicated.

11. Nowadays instead of working with KO versus WT cells one would probably use a complementary degron approach to perturb *Suv39h* and then study the functional consequences. Do the authors observe similar findings when transiently perturbing/depleting *Suv39h*?

Degron and PROTAC technology, that would enable interventions to be targeted specifically to mitosis, will be the next step. Chaetocin (a Suv39h1 inhibitor) was used in the current study to demonstrate the impact of acute/transient perturbation of Suv39h. Chaetocin treatment results in reduced *Esrrb* retention on WT mitotic chromosomes (Figure 4d, Figure 6c, and new figure S4g-h)

Minor comments:

1. For clarity, I would consider rephrasing some of the sentences such as sentence 101-102 ('chromosomes released'), 118-120 ('specifically pairwise deletions of..'),
These have been rephrased in the revised manuscript.

2. What other chromatin modifications are the authors referring to in line 145 ('compensatory increase in H3K27me3 (or other chromatin modifications) rather than loss of H3K9me3 per se..')? Is this referring to DNA methylation?

We meant chromatin modifications such as H3S10ph. Reference to "or other chromatin modifications" has been removed in the revised manuscript for clarity.

3. In figure 3a and d, it seems as if there are 4468 (WT MEFs) and 4519 (Suv39h dn MEFs) proteins detected, rather than the 5789 and 5784 proteins mentioned in the text. Can the authors comment on these differences in number? Also, the numbers displayed in the volcano plots (3a, 3d) on the right (the proteins enriched on sorted chromosomes) do not completely match with the numbers shown in the Venn-diagrams (3b and 3e)

We apologise for any confusion caused. The total protein numbers stated in the manuscript were the number of protein hits detected in the mitotic lysates samples after filtering steps of raw MaxQuant data in Perseus. Volcano plots display proteins that are present in both mitotic lysate and sorted chromosome samples, excluding the protein hits that are present in only one condition (using the robust filtering and statistical analysis method that we have described previously in Djegloul *et al.*, 2020). We have clarified this in the revised narrative and added details on the approach and filtering steps to the materials and methods section. We have also included in supplemental Tables 1 and 2 (tab2) the total number of proteins identified in each condition (mitotic lysates and sorted chromosomes) and number of proteins shared between the lysate and sorted chromosome samples (this number corresponds to the total number of hits displayed in the volcano plots). Finally, the Venn-Diagrams are based on represented protein IDs. A minority of LC-MS/MS hits can correspond to more than one protein ID, which explains the small discrepancy in numbers. We have clarified this in the revised manuscript (in the methods section).

4. It appears as if the black line showing the threshold p-value in figure 4a is not at the same height/cutoff when comparing the left and right volcano plots.

This is the case, as the absolute threshold is not the same for each volcano plot. Instead the threshold is determined for each comparison by a permutation-based FDR.

Perseus plots raw p-values on the y-axis, but significance is based on the t-test q-value (FDR corrected p-value). Additionally, s0 also allows fold changes to play a role (see comment to major point 5), hence the position of threshold can vary between different volcano plots.

5. Can the authors discuss in more detail the factors that are lost from mitotic chromosomes in Suv39h ko cells in their MS experiments?

A full list of these factors is provided in supplemental table 1 tab 3 for the MEFs and supplemental table 2 tab 3 for the ESCs.

6. Are there any non-histone targets of Suv39h that could impact mitotic chromosomes?

Suv39h1 is reported to methylate several non-histone proteins including Rag2, Set8 and Dot1l and the consequences of this have been recently described (reviewed in Weirich *et al.*, 2021). Suv39h1 has also been shown to methylate the mycobacterial histone-like HupB protein (Yaseen *et al.*, 2018), while Suv39h2 can trimethylate LSD1 (Piao *et al.*, 2018), and is reported to methylate lysine 134 of H2A (Sone *et al.*, 2014). As several of these proteins are involved in chromatin regulation, they may be relevant for mitotic chromosomes. We have added a discussion of this to our revised discussion.

Decision Letter, first revision:

Message: Our ref: NSMB-A45961A

22nd Dec 2022

Dear Dr. Fisher,

Thank you for submitting your revised manuscript "Loss of H3K9 tri-methylation alters chromosome compaction and transcription factor retention during mitosis" (NSMB-A45961A). It has now been seen by the original referees and their comments are below. The reviewers find that the paper has improved in revision, and therefore we'll be happy in principle to publish it in Nature Structural & Molecular Biology, pending minor revisions to potentially satisfy the referees' final requests and to comply with our editorial and

formatting guidelines.

We will be performing detailed checks on your paper and will send you a checklist detailing our editorial and formatting requirements. Normally we do this within a week but, as it is the end of the year and most of our personnel is out of the office on vacation, we aim to send you this detailed checklist at the beginning of January. Please do not upload the final materials and make any revisions until you receive this additional information from us.

To facilitate our work at this stage, we would appreciate if you could send us the main text as a word file. Please make sure to copy the NSMB account (cc'ed above).

Sincerely,

Dimitris Typas
Associate Editor
Nature Structural & Molecular Biology
ORCID: 0000-0002-8737-1319

Reviewer #1 (Remarks to the Author):

The authors have addressed all of the concerns raised. This has improved the manuscript, which was already very good. One suggestion the authors may consider is to start their discussion with the broader implications of the work, rather than diving straight into the arguments about Esrrb (although these points are obviously critical).

Reviewer #2 (Remarks to the Author):

The authors have improved their study with additional experiments and edits to the text. Overall, I am satisfied that the main points I raised have been addressed. In particular, the addition of ChIP-qPCR experiments for Esrrb provides evidence that it is indeed mitotic bookmarking of Esrrb which is affected by loss of Suv39h activity. It would have been nice if the authors could have better dissected the transcriptional impact of the loss of H3K9me3-dependent mitotic bookmarking. However, this story makes some interesting observations that provide an opportunity for follow up studies.

Reviewer #3 (Remarks to the Author):

The authors have addressed my main concerns in a satisfactory manner.

Author Rebuttal, first revision:

Reviewer #1:

Remarks to the Author:

The authors have addressed all of the concerns raised. This has improved the manuscript, which was already very good. One suggestion the authors may consider is to start their discussion with the broader implications of the work, rather than diving straight into the arguments about Esrrb (although these points are obviously critical).

Reviewer #2:

Remarks to the Author:

The authors have improved their study with additional experiments and edits to the text. Overall, I am satisfied that the main points I raised have been addressed. In particular, the addition of ChIP-qPCR experiments for Esrrb provides evidence that it is indeed mitotic bookmarking of Esrrb which is affected by loss of Suv39h activity. It would have been nice if the authors could have better dissected the transcriptional impact of the loss of H3K9me3-dependent mitotic bookmarking. However, this story makes some interesting observations that provide an opportunity for follow up studies.

Reviewer #3:

Remarks to the Author:

The authors have addressed my main concerns in a satisfactory manner

Response

We are grateful to each of the reviewers for their positive comments. We have reorganised the discussion section as requested by reviewer 1, so that we start a discussion of the broader impacts of our work ahead of the discussing changes in Esrrb retention.

Final Decision Letter:

Message 13th Feb 2023

:

Dear Dr. Fisher,

We are now happy to accept your revised paper "Loss of H3K9 tri-methylation alters chromosome compaction and transcription factor retention during mitosis" for publication as a Article in Nature Structural & Molecular Biology.

As soon as your article is published, you can generate your shareable link by entering the DOI of your article here: http://authors.springernature.com/share. Corresponding authors will also receive an automated email with the shareable link

Your paper will be published online soon after we receive proof corrections and will appear in print in the next available issue. You can find out your date of online publication by contacting the production team shortly after sending your proof corrections. Content is published online weekly on Mondays and Thursdays, and the embargo is set at 16:00 London time (GMT)/11:00 am US Eastern time (EST) on the day of publication. Now is the time to inform your Public Relations or Press Office about your paper, as they might be interested in promoting its publication. This will allow them time to prepare an accurate and satisfactory press release. Include your manuscript tracking number (NSMB-A45961B) and our journal name, which they will need when they contact our press office.

About one week before your paper is published online, we shall be distributing a press release to news organizations worldwide, which may very well include details of your work. We are happy for your institution or funding agency to prepare its own press release, but it

must mention the embargo date and Nature Structural & Molecular Biology. If you or your Press Office have any enquiries in the meantime, please contact press@nature.com.

Please note that *Nature Structural & Molecular Biology* is a Transformative Journal (TJ). Authors may publish their research with us through the traditional subscription access route or make their paper immediately open access through payment of an article-processing charge (APC). Authors will not be required to make a final decision about access to their article until it has been accepted. Find out more about Transformative Journals <https://www.springernature.com/gp/open-research/transformative-journals>

You will not receive your proofs until the publishing agreement has been received through

our system.

Sincerely,

Dimitris Typas
Associate Editor
Nature Structural & Molecular Biology
ORCID: 0000-0002-8737-1319

Click here if you would like to recommend Nature Structural & Molecular Biology to your librarian:

<http://www.nature.com/subscriptions/recommend.html#forms>